# Nox4 regulates InsP₃ receptor-dependent Ca²⁺ release into mitochondria to promote cell survival

Matteo Beretta[1], Celio XC Santos[1], Chris Molenaar[1], Anne D Hafstad[1,2], Chris CJ Miller[3],
Aram Revazian[4], Kai Betteridge[1], Katrin Schröder[5], Katrin Streckfuß-Bömeke[6],
James H Doroshow[7], Roland A Fleck[8], Tsung-Ping Su[9], Vsevolod V Belousov[4,10,11],
Maddy Parsons[12] & Ajay M Shah[1,*]

## Abstract

Cells subjected to environmental stresses undergo regulated cell death (RCD) when homeostatic programs fail to maintain viability. A major mechanism of RCD is the excessive calcium loading of mitochondria and consequent triggering of the mitochondrial permeability transition (mPT), which is especially important in post-mitotic cells such as cardiomyocytes and neurons. Here, we show that stress-induced upregulation of the ROS-generating protein Nox4 at the ER-mitochondria contact sites (MAMs) is a pro-survival mechanism that inhibits calcium transfer through InsP₃ receptors (InsP₃R). Nox4 mediates redox signaling at the MAM of stressed cells to augment Akt-dependent phosphorylation of InsP₃R, thereby inhibiting calcium flux and mPT-dependent necrosis. In hearts subjected to ischemia–reperfusion, Nox4 limits infarct size through this mechanism. These results uncover a hitherto unrecognized stress pathway, whereby a ROS-generating protein mediates pro-survival effects through spatially confined signaling at the MAM to regulate ER to mitochondria calcium flux and triggering of the mPT.

**Keywords** calcium signaling; cell death; InsP₃ receptor; mitochondria-associated membrane; NADPH oxidase-4
**Subject Categories** Autophagy & Cell Death; Membrane & Trafficking
**The EMBO Journal (2020) 39: e103530**

## Introduction

In response to significant extracellular or intracellular environmental perturbations, eukaryotic cells activate diverse stress pathways designed to restore homeostasis and maintain cell viability. These include pathways such as the integrated stress response, nutrient sensing response, electrophilic stress response, DNA damage response, and many others. When such adaptive pathways fail to restore homeostasis in the face of sustained or severe stress, cells may undergo regulated cell death (RCD)—an act of active cellular suicide that is considered to be important in maintaining the integrity of the organism as a whole (Galluzzi *et al*, 2014). The balance between homeostatic stress responses and RCD is especially important for the fate of post-mitotic cells, such as cardiomyocytes and neurons, which have very limited regenerative capacity (Whelan *et al*, 2010). RCD may occur through different signaling pathways depending upon the stress, physiologic/pathologic setting, and cell type, and results in distinct morphological and functional phenotypes —e.g., apoptosis or necrosis. Although apoptosis depends upon the activation of caspase cascades triggered either by cell surface receptors or cell-intrinsic mechanisms, regulated necrosis is a caspase-independent molecular signaling process (Galluzzi *et al*, 2014).

Mitochondrial calcium overload is a major mechanism of RCD through the triggering of the mitochondrial permeability transition (mPT; Giorgi *et al*, 2018). This involves an abrupt increase in the conductance of the mPT pore complex (mPTP) at the inner mitochondrial membrane, resulting in free entry of solutes, dissipation of the mitochondrial membrane potential ($\Delta\Psi_m$), loss of oxidative

---

1  School of Cardiovascular Medicine & Sciences, King's College London British Heart Foundation Centre, London, UK
2  Cardiovascular Research Group, Department of Medical Biology, UIT-The Arctic University of Norway, Tromsø, Norway
3  Department of Basic and Clinical Neuroscience, Institute of Psychiatry, Psychology and Neuroscience, King's College London, London, UK
4  Institute for Cardiovascular Physiology, Georg August University Göttingen, Göttingen, Germany
5  Institute for Cardiovascular Physiology, Goethe-University Frankfurt, Frankfurt am Main, Germany
6  Department of Cardiology and Pneumology, Universitätsmedizin Göttingen, Göttingen, Germany
7  Division of Cancer Treatment and Diagnosis, National Cancer Institute, NIH, Bethesda, MD, USA
8  Centre for Ultrastructural Imaging, King's College London, London, UK
9  Cellular Pathobiology Section, National Institute on Drug Abuse, NIH, Baltimore, MD, USA
10 Shemyakin-Ovchinnikov Institute of Bioorganic Chemistry, Moscow, Russia
11 Pirogov Russian National Research Medical University, Moscow, Russia
12 King's College London British Heart Foundation Centre, Randall Centre of Cell and Molecular Biophysics, London, UK
   *Corresponding author. Tel: +44 207848 5189; E-mail: ajay.shah@kcl.ac.uk

phosphorylation-dependent ATP generation, mitochondrial swelling and rupture, and eventual cell death (Whelan *et al*, 2010; Izzo *et al*, 2016). Increased calcium influx from endoplasmic reticulum (ER) to mitochondria leading to mPT is a well-established death pathway that is important in many somatic and cancer cells (Marchi *et al*, 2018). The major channels for calcium release from the ER are inositol 1,4,5-trisphosphate ($InsP_3$) receptors ($InsP_3R$), while calcium uptake into the mitochondria is mediated by voltage-dependent anion channels (VDAC) at the outer mitochondrial membrane and the mitochondrial calcium uniporter complex (MCU) at the inner mitochondrial membrane (De Stefani *et al*, 2011). Calcium overload-associated mPT is also centrally involved in cell death during ischemia–reperfusion injury of the heart or brain, in this case driving regulated necrosis and contributing to myocardial infarction and stroke, respectively (Whelan *et al*, 2010; Fuchs & Steller, 2011; Kung *et al*, 2011). The maintenance of mitochondrial calcium homeostasis is therefore critical in the determination of cell fate.

NADPH oxidase (Nox) proteins are a family of complex multi-subunit enzymes that catalyze electron transfer from NADPH to molecular $O_2$ and generate ROS as their primary function. Noxs are especially important in localized redox signaling and play key roles in physiologic and pathologic processes such as cell differentiation, migration, proliferation, and tissue remodeling (Lassegue *et al*, 2012). Among the seven mammalian Noxs (Nox1-5 and Duox1-2), Nox4 has the most widespread tissue distribution and has been found to mediate protective effects during cellular stress situations (Zhang *et al*, 2010; Nlandu Khodo *et al*, 2012; Schroder *et al*, 2012), in contrast to most other Noxs which drive detrimental tissue remodeling (Lassegue *et al*, 2012). Notably, Nox4 was found to be protective against cell death in the brain (Basuroy *et al*, 2011), lung (Carnesecchi *et al*, 2011), retina (Groeger *et al*, 2009), pancreas (Lee *et al*, 2007), heart (Santos *et al*, 2016), and several types of cancer (Zhang *et al*, 2013; Zhu *et al*, 2013). However, the mechanism of this pro-survival effect in multiple cell types is unclear.

Here, we report that Nox4 mediates a homeostatic survival response by inhibiting calcium transfer to mitochondria through $InsP_3$ channels and preventing RCD. We find that Nox4 is localized at mitochondria-associated membrane or MAM, sites of physical membrane contact between the ER and mitochondria and key hubs for the regulation of mitochondrial calcium levels (Rizzuto *et al*, 1998; Hajnoczky *et al*, 2002; Filadi *et al*, 2017). During cellular stress, Nox4 is upregulated at the MAM where it promotes Akt-mediated phosphorylation of $InsP_3R$ to block calcium transfer and downstream mPT-dependent cell death. This action of Nox4 involves highly localized redox signaling at the MAM, distinct from the RCD-promoting effects known to be associated with excessive ROS generation by other sources. Nox4 is robustly protective against mPT-dependent necrosis in cells as well as in hearts subjected to ischemia–reperfusion injury. Collectively, these findings identify a previously unrecognized homeostatic pro-survival pathway in which spatially confined Nox4-dependent ROS production and signaling at the MAM inhibits a centrally important pathway of RCD.

## Results

### Nox4 protects cells against mPT-dependent death

Serum starvation is a well-established stress stimulus that may lead to RCD (Kulkarni & McCulloch, 1994). We first investigated the effects of serum starvation of cultured cardiac myocytes on Nox4 levels. The serum deprivation of cells resulted in a progressive increase in Nox4 mRNA and protein levels as compared to serum-replete cells (Fig 1A and B, and Appendix Fig S1A). To assess the role of Nox4 in this setting, we depleted Nox4 and then quantified cell death in response to 48 h of serum starvation. Cells with adenoviral shRNA-mediated depletion of Nox4 showed a significantly higher level of cell death than those infected with a control shRNA, across a range of different degrees of serum starvation (Fig 1C).

We assessed the type of cell death that was enhanced in serum-starved Nox4-depleted cells. We found that the release of LDH and high mobility group box 1 (HMGB1) protein were significantly increased in Nox4-deficient compared to control cells (Fig 1D and E). However, there were no differences between Nox4-deficient and

---

**Figure 1. Nox4 inhibits mPT-dependent regulated cell death.**

A, B   Quantification of Nox4 mRNA levels (A) and protein levels (B) in cultured rat cardiomyocytes after increasing degrees of serum starvation as compared to serum-replete cells (15% serum). *n* = 4 cell preparations/group. A representative immunoblot for Nox4 and tubulin as a loading control is shown in panel B.

C   Cell death after 48 h serum starvation in cardiomyocytes in which Nox4 was depleted by an adenoviral shRNA (Ad.shNox4) as compared to cells infected with a control vector (Ad.Ctl). *n* = 6/group. The representative immunoblot shows reduction in Nox4 protein levels after shRNA-mediated knockdown.

D   Lactate dehydrogenase (LDH) activity in cell supernatants. Staurosporine (Stauro, 1 μmol/l) was used to induce cell death by apoptosis. *n* = 4 cell preparations/group.

E   Levels of high mobility group box 1 protein (HMGB1) in cell supernatants. *n* = 3/group. A representative immunoblot is shown at the top.

F   Protein levels of cytochrome *c* in the cytoplasmic fractions of cardiomyocytes. *n* = 4/group. A representative immunoblot is shown at the top. The cellular membrane fraction (Mem) containing mitochondria was used as a positive control.

G   Percentage of Nox4-depleted and control cells with depolarized mitochondria, quantified by flow cytometry. *n* = 3 cell preparations/group (100,000 cells per experiment).

H   Effect of cyclosporin A (CsA, 1 μmol/l) on the percentage of cells with depolarized mitochondria. *n* = 6–8/group.

I   Effect of cyclosporin A (CsA, 1 μmol/l) on cell death in serum-starved cardiomyocytes. *n* = 4/group.

J   Cell death in serum-starved Nox4 knockout MEFs (Nox4KO) and wild-type MEFs (WT). Cat = PEG-catalase (500 U/ml). Nox4KO MEFs were transfected either with active Nox4 or a catalytic inactive Nox4 mutant, Nox4[P437H] (Mut). *n* = 6–12/group.

K   HMGB1 levels in the supernatants of serum-starved MEFs in similar experiments to those in (J). *n* = 4/group. A representative immunoblot is shown to the top.

L   Percentage of MEFs with depolarized mitochondria. *n* = 6/group.

Data information: Data are mean ± SEM. *$P < 0.05$; **$P < 0.01$; ***$P < 0.001$; ****$P < 0.0001$ among compared groups or vs. control. [#]$P < 0.05$; [####]$P < 0.0001$ vs. Nox4KO. (C and G), 2-way ANOVA; all other panels, 1-way ANOVA.

Source data are available online for this figure.

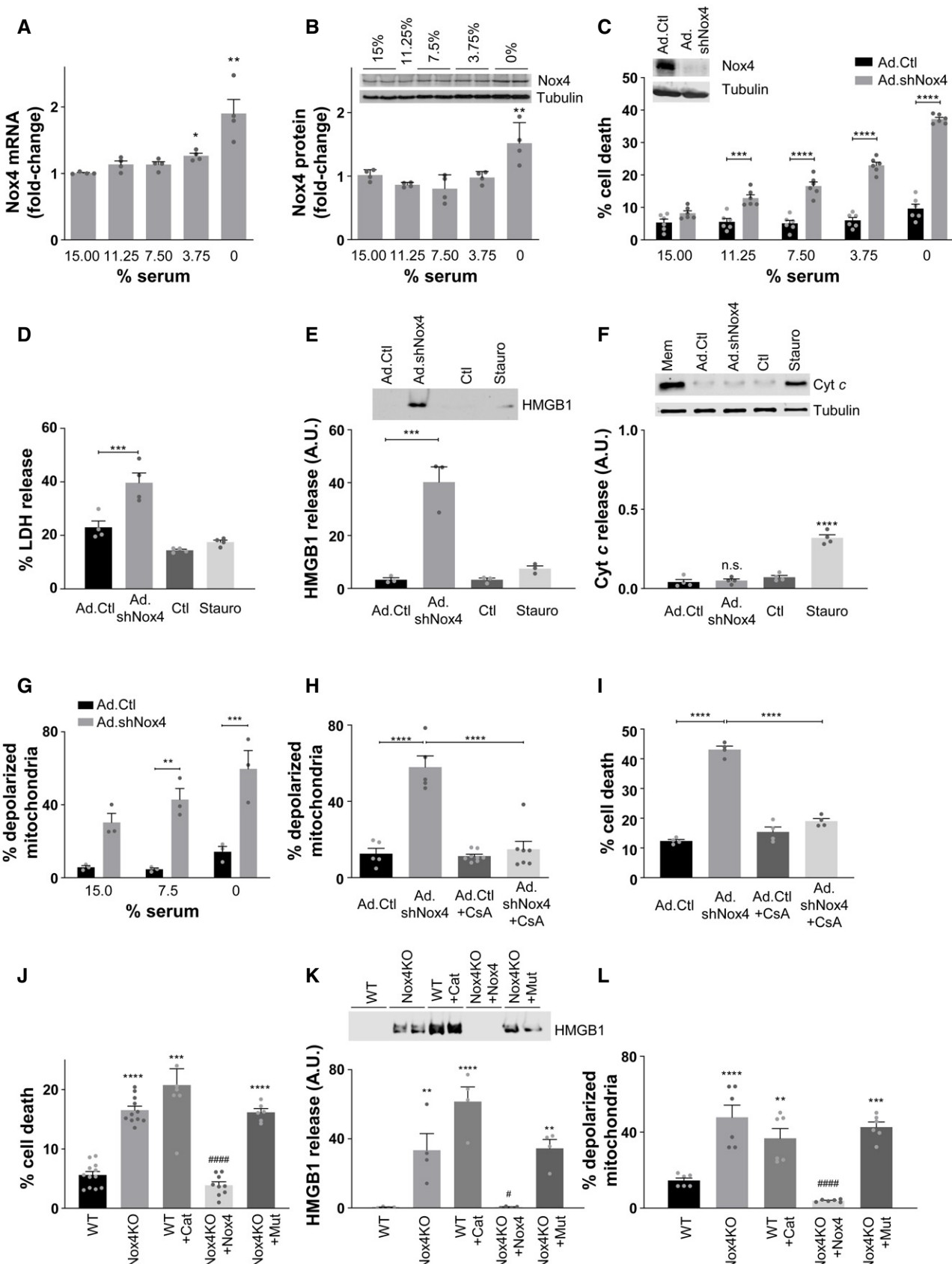

Figure 1.

control cells in the levels of cleaved caspase-3 or caspase-8 (Appendix Fig S1B and C) nor in cytochrome $c$ release to the cytoplasm (Fig 1F). The protein kinase inhibitor staurosporine was used as a positive control for the induction of apoptosis in these experiments. These results suggest that the mode of death in serum-starved Nox4-deficient cells is necrosis rather than apoptosis. Regulated necrosis may involve a variety of different mechanisms including the involvement of receptor-interacting protein kinase 1 (RIPK1), RIPK3, polyADP-ribose polymerase 1 (PARP), or apoptosis-inducing factor (AIF; Galluzzi et al, 2014). However, none of these pathways appeared to be activated to a greater extent in Nox4-deficient compared to control cells (Appendix Fig S1D). An alternative pathway of regulated necrosis involves the dissipation of $\Delta\Psi_m$ and induction of the mPT. To assess whether the mPT is involved in the cell death of Nox4-deficient cells, we measured $\Delta\Psi_m$ in these cells after serum starvation. We found that there was an > 4-fold increase in the proportion of cells with depolarized mitochondria under Nox4-deficient conditions (Fig 1G). Following treatment with cyclosporin A, which inhibits cyclophilin D—a component of the mPTP complex that is required for pore opening (Baines et al, 2005) —the proportion of depolarized mitochondria and cell death in Nox4-deficient cells was reduced to a similar level to that in control cells (Fig 1H and I). A different inhibitor of the mPTP, bongkrekic acid, also reduced cell death in Nox4-deficient cells (Appendix Fig S1E). These data suggest that Nox4 deficiency leads to an increase in mPT-dependent regulated necrosis during serum starvation.

To assess whether Nox4-dependent regulation of necrosis also occurs in other cell types, we studied mouse embryonic fibroblasts (MEFs) from Nox4 knockout (Nox4 KO) animals and matched wild-type (WT) MEFs. After serum starvation for 48 h, the level of cell death was ~ 3-fold higher in Nox4 KO MEFs than WT MEFs (Fig 1J). This was accompanied by an increase in HMGB1 release (Fig 1K) and an increased proportion of cells with depolarized mitochondria (Fig 1L), but no changes in cytochrome c release to the cytoplasm (Appendix Fig S1F)—indicating that Nox4 inhibits mPT-dependent RCD also in these cells. Since Nox4 generates ROS (hydrogen peroxide, $H_2O_2$), we tested whether its enzymatic activity and ROS generation were required for the inhibition of mPT-dependent RCD. The re-introduction of Nox4 into KO MEFs (Appendix Fig S1G) markedly reduced cell death, HMGB1 release, and mitochondrial

depolarization under serum starvation (Fig 1J–L). However, the overexpression of a catalytically inactive mutant of Nox4 which has a proline to histidine amino acid substitution at residue 437 in the NADPH-binding domain (Nox4[P437H]; Dinauer et al, 1989; Appendix Fig S1G) failed to rescue mitochondrial depolarization, HMGB1 release, and cell death in KO MEFs (Fig 1J–L). On the other hand, the treatment of WT MEFs with PEG-catalase to degrade hydrogen peroxide resulted in a significant increase in mitochondrial depolarization, HMGB1 release, and cell death during serum starvation (Fig 1J–L). Taken together, these results indicate that endogenous Nox4 is upregulated during serum starvation of cells and inhibits mPT-dependent RCD through effects that involve ROS production.

## Mitochondrial calcium levels are increased in Nox4-deficient cells

To investigate the mechanism underlying Nox4-mediated regulation of mPT-dependent RCD, we started by assessing mitochondrial calcium levels since mitochondrial calcium overload is a key trigger of the mPT. In cultured cardiomyocytes in which Nox4 levels were depleted using two different siRNAs and then subjected to serum starvation, mitochondrial calcium levels were significantly increased (by ~10%) compared to control cells (Fig 2A and Appendix Fig S2A). Similarly, Nox4 KO MEFs had significantly higher mitochondrial calcium levels than WT MEFs (Fig 2B and Appendix Fig S2B). To study calcium transfer to mitochondria, we treated cells with either histamine or ATP to induce calcium release from ER stores and then assessed mitochondrial, cytosolic, and ER calcium levels using targeted probes. Following treatment with either agonist, mitochondrial calcium levels were substantially higher in serum-starved Nox4 KO compared to WT cells (Figs 2C and D, and EV1A and B). ER calcium levels were marginally lower in Nox4 KO compared to WT cells after histamine (Figs 2E and EV1C), with no significant difference in cytosolic calcium transients between groups (Appendix Fig S2C and Fig EV1E). After treatment with ATP, the difference in ER calcium between Nox4 KO and WT cells was greater (~8%) (Figs 2F and EV1D) but cytosolic calcium transients were similar between groups (Appendix Fig S2D and Fig EV1F). The results with histamine are consistent with enhanced calcium

---

**Figure 2. Nox4 regulates calcium transfer to mitochondria.**

A Basal mitochondrial calcium levels assessed in serum-starved rat cardiomyocytes using a mitochondrial-targeted cameleon CFP/YFP FRET probe. Nox4 was depleted with silencing RNAs (siRNAs) or cells were transfected with a control scrambled siRNA (siScr). Representative photomicrographs are shown at the top. The spectrum color scale represents the ratio of emitted fluorescence (YFP/CFP). The mean changes in Nox4-depleted cells as compared to scrambled control (dotted line) are shown at the bottom. The representative immunoblot shows depletion of Nox4 protein levels. Scale bars: 10 μm. $n$ = 3/group (with > 50 cells per individual experiment).

B Basal mitochondrial calcium levels in WT and Nox4KO MEFs after serum starvation. Representative photomicrographs are shown at the top. The spectrum color scale represents the ratio of emitted fluorescence (YFP/CFP). Mean data at the bottom. Scale bars: 10 μm. $n$ = 3/group (with > 100 cells per individual experiment).

C, D Changes in mitochondrial calcium levels in serum-starved WT and Nox4KO MEFs after the addition of histamine (100 μmol/l, C) or ATP (100 μmol/l, D). $n$ = 3/group (with > 30 cells per individual experiment).

E, F Changes in ER calcium levels measured with an ER-targeted cameleon probe in serum-starved WT and Nox4KO MEFs after the addition of histamine (100 μmol/l, E) or ATP (100 μmol/l, F). $n$ = 3/group (with > 30 cells per individual experiment).

G Peak increase in mitochondrial calcium levels response in response to histamine (100 μmol/l) in WT MEFs with or without treatment with PEG-catalase (Cat), and in Nox4KO MEFs with or without transfection with active Nox4 or a catalytically inactive Nox4[P437H] mutant (Mut). $n$ = 3/group (with > 30 cells per individual experiment).

H Representative time course of histamine-induced changes in mitochondrial calcium for the experiments shown in (G).

Data information: Data are mean ± SEM. **$P$ < 0.01; ****$P$ < 0.0001 among compared groups or vs. control. ####$P$ < 0.0001 vs. Nox4KO. (A, B): Student's $t$-test; (C–H): 2-way repeated measures ANOVA; (I): 1-way ANOVA.

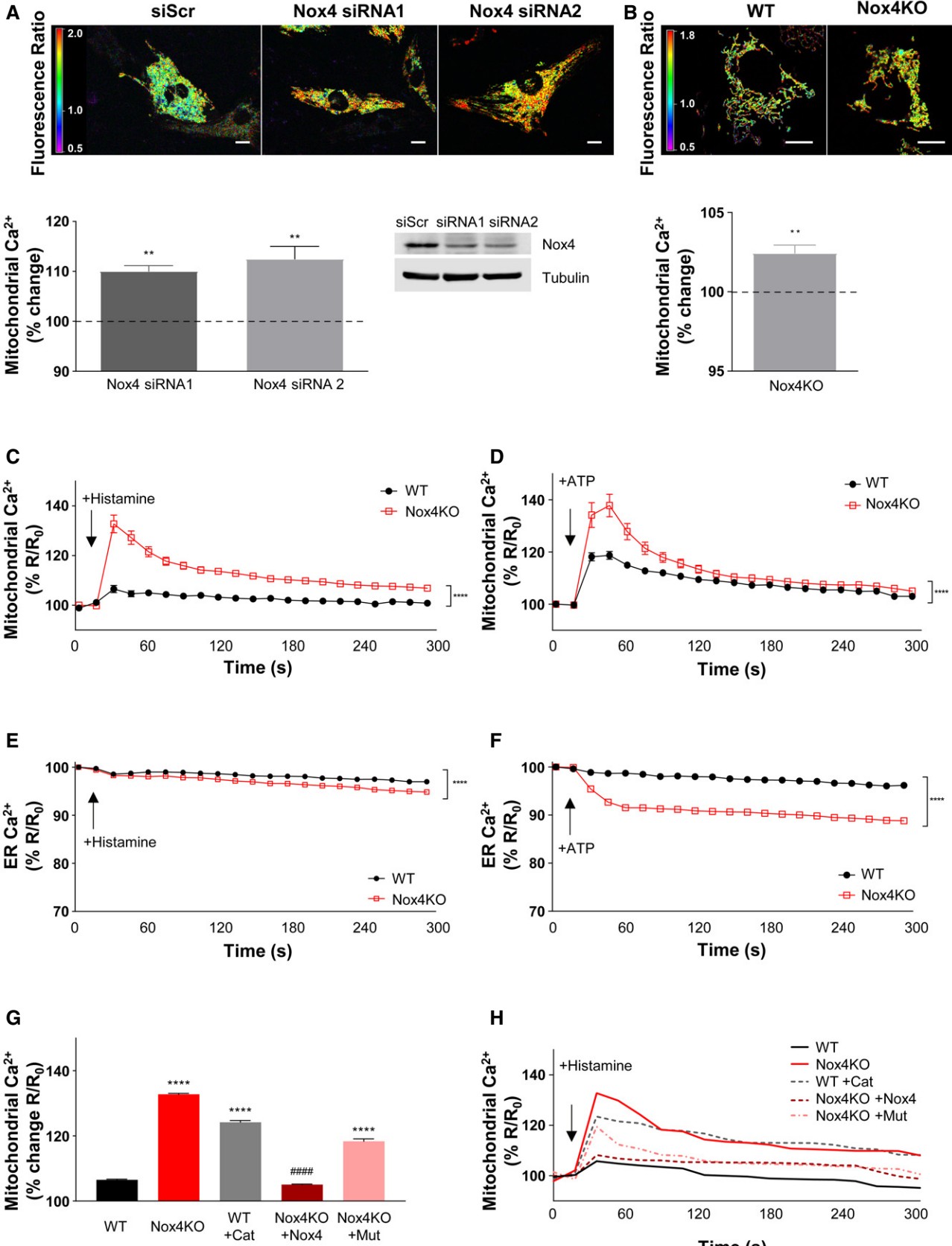

Figure 2.

transfer from the ER to mitochondria in the Nox4 KO cells. In the case of ATP, the greater depletion of ER calcium in Nox4 KO versus WT cells suggests that there may be additional effects. This magnitude of difference in ER calcium would be expected to be accompanied by differences in cytosolic calcium transients, so there may be other compensatory mechanisms at play with this agonist. To confirm the Nox4 dependence of these effects, we transfected Nox4 KO cells with either Nox4 or Nox4$^{P437H}$ and then quantified mitochondrial calcium levels after histamine stimulation under serum starvation conditions. This analysis showed that the re-introduction of Nox4 restored mitochondrial calcium levels of KO cells to the level observed in WT cells but that Nox4$^{P437H}$ was ineffective (Figs 2G and H, and EV1G). We also found that the treatment of serum-starved WT cells with PEG-catalase resulted in a marked increase in mitochondrial calcium levels after histamine stimulation (Figs 2G and H, and EV1G). No difference in mitochondrial calcium levels was found between histamine-stimulated WT and Nox4 KO cells under serum-replete conditions (Appendix Fig S2E).

The uptake of ER-released calcium by mitochondria requires them to be energized and strictly depends upon the $\Delta\Psi_m$, whereas depolarized mitochondria could become calcium overloaded through other routes (e.g., via the mPTP; Giorgi et al, 2018). To distinguish between these possibilities, we assessed the changes in mitochondrial calcium upon histamine stimulation in cells loaded with tetramethylrhodamine ethyl ester (TMRE) to index $\Delta\Psi_m$. We found that only TMRE$^+$ cells showed an enhanced mitochondrial calcium influx in Nox4-deficient compared to WT conditions, whereas calcium levels were similar in TMRE$^-$ Nox4 KO and WT cells (Fig EV2A and B). Furthermore, the higher calcium influx in TMRE$^+$ Nox4-deficient cells was unaffected by inhibition of the mPTP (Fig EV2C). In cardiomyocytes, a major pathway for calcium release from the ER (or sarcoplasmic reticulum [SR]) is via ryanodine receptors, in addition to InsP$_3$R-mediated release. To assess the relative contribution of these pathways to enhanced mitochondrial calcium uptake under serum-deficient conditions, we compared the responses to caffeine (which releases calcium via ryanodine receptors) and histamine in cardiomyocytes. Both agonists robustly induced calcium release into the cytosol but an enhanced mitochondrial calcium uptake under Nox4-deficient conditions was observed only with histamine (Fig EV2D).

Taken together, these data suggest that Nox4 regulates ER to mitochondrial calcium transfer via InsP$_3$R during serum starvation, with the level of transfer and subsequent mitochondrial calcium levels being much higher when Nox4 is deficient or inactive.

## Nox4 is localized at the MAM

We surmised that the intracellular localization of Nox4 during stress may be critical in mediating the above responses. Many previous studies have reported an ER localization of Nox4 (Ambasta et al, 2004; Anilkumar et al, 2008) but some studies also suggested a mitochondrial localization (Block et al, 2009) although definitive biochemical evidence for the latter suggestion through careful subcellular fractionation is lacking. In view of the effect on calcium transfer to mitochondria and the fact that the MAM is a hotspot for InsP$_3$R-mediated calcium release from the ER into mitochondria (Csordas et al, 1999), this was a location of particular interest. We undertook fractionation studies to isolate cytosol, ER, crude mitochondrial, pure mitochondrial, and MAM fractions from cardiac cells. Using a well-validated polyclonal Nox4 antibody, we found that Nox4 was highly enriched in the ER and MAM fractions (Fig 3A). The MAM fraction also contained the established MAM-enriched proteins VDAC, Sigma1 receptor, FACL4, and calnexin. We undertook subcellular fractionation in whole heart and kidney tissue and found a similar localization of Nox4 (Fig 3B and C). Nox4 levels at the MAM were significantly higher under serum starvation compared to serum-replete conditions (Fig EV3A).

To obtain complementary evidence of Nox4 localization, we undertook electron microscopy studies. In mouse heart sections, we observed a localization of both Nox4 and the MAM marker FACL4 in close contact with the ER cisternae in the proximity of mitochondria, indicative of a MAM localization (Figs 3D and Appendix S3A). No Nox4 immunolabeling was observed in heart tissue from Nox4 KO mice. We used a well-characterized anti-human Nox4 monoclonal antibody (Meitzler et al, 2017) to perform studies in human-induced pluripotent stem cell-derived cardiomyocytes (hiPSC-CM) and obtained similar results in these cells (Fig 3D and Appendix Fig S3B). Lentiviral-mediated knockdown of Nox4 abolished Nox4 immunolabeling in the cardiomyocytes, confirming antibody specificity.

As a third independent approach, we performed confocal microscopy in cells. Initial confocal microscopy in rat cardiomyocytes indicated a co-localization of Nox4 with either InsP$_3$R or FACL4 (Appendix Fig S4A). We then undertook in situ proximity ligation studies in rat cardiomyocytes, WT and Nox4 KO MEFs, and hiPSC-CM to detect spatial proximity (within 30–40 nm) of relevant proteins. These experiments showed that Nox4 was in close proximity to FACL4 and InsP$_3$R in WT MEFs, whereas no co-localization was observed in KO cells (Fig 4A, and Appendix Fig S4B and E).

**Figure 3. Nox4 is localized at the mitochondria-associated ER membrane (MAM).**

A  Immunoblotting of subcellular fractions from serum-starved rat cardiomyoblasts (H9c2 cells). Input: homogenate before fractionation; Cyto: cytosol; ER: endoplasmic reticulum; Crude M: crude mitochondrial fraction, also containing MAM; Pure Mito: mitochondria; MAM: mitochondria-associated ER membranes. Cytochrome c (Cyt c) was used as a mitochondrial marker, VDAC as an outer mitochondrial membrane marker, calnexin as an ER marker, Sigma1R and FACL4 as MAM markers, and tubulin as a cytosolic marker.
B  Immunoblotting of subcellular fractions from rat heart.
C  Immunoblotting of subcellular fractions from rat kidney.
D  Electron micrographs showing subcellular localization of Nox4 in wild-type (WT) and Nox4KO mouse hearts, and human-induced pluripotent stem cell-derived cardiomyocytes (hiPSC-CM). hiPSC-CM were depleted of Nox4 using a lentiviral shRNA (Nox4KD) or infected with a control lentivirus. FACL4 was used as a MAM marker. Arrowheads show peri-mitochondrial localization of Nox4 and FACL4, in the proximity of ER cisternae and ribosomes. M = mitochondria, ER = endoplasmic reticulum, T = T-tubule. Scale bars: 100 nm.

Source data are available online for this figure.

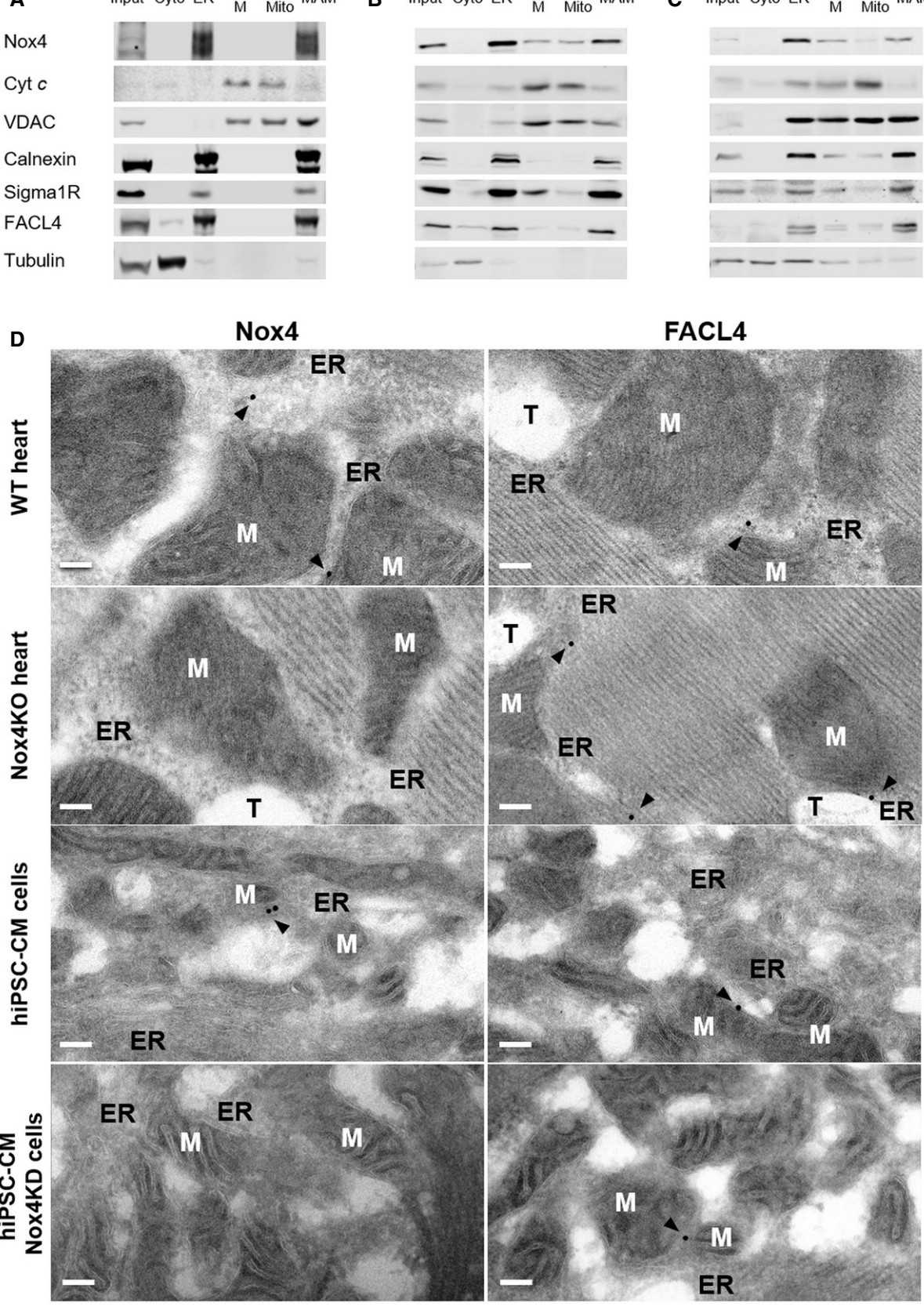

Figure 3.

The absence of Nox4 did not affect the co-localization of FACL4 and InsP$_3$R. As another control, we co-stained for Nox4 and a lysosome marker, LAMP1, but found no evidence of co-localization in this case (Appendix Fig S4B and E). The co-localization of Nox4 and FACL4 was significantly higher under serum starvation than under serum-replete conditions (Fig EV3B). A co-localization of Nox4, FACL4, and InsP$_3$R was also observed in rat cardiomyocytes and hiPSC-CM, whereas the signal was absent in cells in which Nox4 was depleted by siRNA- or shRNA-mediated knockdown (Fig 4B and C, and Appendix Fig S4C, D, F and G). After expression of Nox4 or Nox4$^{P437H}$ in Nox4 KO cells, the co-localization with FACL4 and InsP$_3$R was restored but there was no change in the FACL4/InsP$_3$R signal suggesting that Nox4 does not alter MAM formation *per se* (Appendix Fig S4H). Interestingly, the number of interactions (dots/cell) appeared to be much higher in the cardiac cells than MEFs, perhaps related to the higher mitochondrial density in these cells. Collectively, these experiments using 3 complementary approaches provide strong evidence that Nox4 has a localization at the MAM, the domain of close and dynamic interaction between the ER and mitochondria.

### Nox4 augments Akt-dependent InsP$_3$ receptor phosphorylation

We wanted to ascertain how MAM-localized Nox4 regulates calcium transfer to mitochondria. Previous studies reported that Nox4 can enhance Akt (protein kinase B) activation (Mahadev *et al*, 2004; Anilkumar *et al*, 2008), while Akt-dependent phosphorylation of InsP$_3$R was shown to be a mechanism that diminishes ER calcium release, including under conditions of serum deprivation (Szado *et al*, 2008; Giorgi *et al*, 2010; Marchi *et al*, 2012). Analysis of the subcellular fractions obtained from cardiac cells (Fig 5A) confirmed that the MAM fraction was enriched in InsP$_3$R, Akt, and protein phosphatase 2a (PP2a, which modulates the extent of Akt phosphorylation; Giorgi *et al*, 2010). To assess the potential co-localization of proteins in a signaling complex at the MAM, we subjected the crude mitochondrial fraction of cells (i.e., the fraction containing mitochondria and MAM) to sucrose gradient centrifugation, a method that can separate macromolecular complexes based on their size and shape. This analysis revealed that Nox4, PP2a, Akt, and InsP$_3$R co-eluted in the same fractions (#10-12), along with the MAM marker FACL4 (Fig 5B). We next assessed the activation of Akt and the phosphorylation of InsP$_3$R at its Akt substrate motif R*X*R*XX*(S/T) in WT versus Nox4 KO cell fractions after serum starvation. InsP$_3$R phosphorylation at the Akt site was detected by first

immunoprecipitating InsP$_3$R and then immunoblotting with an antibody against the phosphorylated Akt substrate motif (Khan *et al*, 2006; Appendix Fig S5A). We found that both Akt phosphorylation (activation) and InsP$_3$ phosphorylation were significantly lower in Nox4 KO cells compared to WT (Fig 5C). There were no significant differences between the cells under serum-replete conditions (Appendix Fig S5B). The re-introduction of Nox4 into Nox4 KO cells augmented both Akt activation and InsP$_3$R phosphorylation during serum starvation, whereas the inactive Nox4$^{P437H}$ mutant was without effect (Fig 5C). We also found that the pre-incubation of serum-starved WT cells with PEG-catalase significantly reduced both Akt and InsP$_3$R phosphorylation. Therefore, Nox4-dependent enhancement of Akt activation and InsP$_3$R phosphorylation requires Nox4 enzymatic activity and ROS production.

To validate Nox4-dependent ROS production and its subcellular localization under these conditions, we used targeted probes that allow the imaging of H$_2$O$_2$ in different subcellular compartments. We engineered one probe to target it to the MAM by fusing it to FACL4, whereas a second probe indexed cytosolic H$_2$O$_2$ levels. We observed a significantly higher ROS signal at the MAM in WT cells compared to Nox4 KO cells during serum starvation but no difference in cytosolic ROS between the groups (Fig 5D). A higher MAM-located ROS signal in WT versus Nox4 KO cells was also found under serum-replete conditions but the magnitude was lower than under serum-deficient conditions (Appendix Fig S5C). The MAM-located ROS signal was restored in Nox4 KO cells after the transfection of functional Nox4 but not by Nox4$^{P437H}$ (Fig 5D). Mutant redox-insensitive ROS probes were used to control for any pH-dependent changes in these experiments (Appendix Fig S5D). No difference in ROS signal between WT and Nox4 KO cells was observed when a mitochondrial-targeted probe was used (Fig EV4A). Furthermore, the incubation of cells with a mitochondrial-targeted antioxidant, Mito-TEMPO, during serum starvation did not alter InsP$_3$R phosphorylation (Fig EV4B). Incubation with Mito-TEMPO also had no effect on the histamine-induced increase in mitochondrial calcium (Fig EV4C). These results indicate that mitochondrial ROS do not appear to be involved in the changes in InsP$_3$R phosphorylation and mitochondrial calcium.

A likely mechanism for Nox4/ROS-dependent augmentation of Akt activation may be the inhibition of PP2a activity, which is known to be redox-sensitive (Foley *et al*, 2007). Consistent with this possibility, we found that the serine/threonine phosphatase activity in crude mitochondrial fractions of serum-starved WT cells was ~2-fold lower than in Nox4 KO cells, with the major component of this

---

**Figure 4. In situ proximity ligation of Nox4 and MAM markers.**

A  Simplified photomicrographs of proximity ligation studies in WT and Nox4KO MEFs, showing cell borders, nuclei (blue), and yellow dots corresponding to co-localization of proteins. Quantification of the number of dots/cell in each condition is shown to the right. Proximity was tested for the following protein couples: FACL4/Nox4, InsP$_3$R/Nox4, and FACL4/InsP$_3$R. Scale bars: 10 µm. *n* = 3/group (with > 30 cells/individual experiment). Controls are shown in Appendix Fig S4B and the original unsimplified photographs in Appendix Fig S4E.

B  Simplified photomicrographs of proximity ligation staining in rat cardiomyocytes transfected with Nox4 siRNA or a scrambled control (siScr), with quantification as in (A). Proximity was tested for the FACL4/Nox4 protein couple. Scale bars: 10 µm. *n* = 3/group (with > 30 cells/individual experiment). Controls and other protein couples are shown in Appendix Fig S4C and the original unsimplified photographs in Appendix Fig S4F.

C  Simplified photomicrographs of proximity ligation staining in hiPSC-CM (human cardiomyocytes) transduced with Nox4 shRNA or scrambled control, with quantification as in (A). Proximity was tested for the FACL4/Nox4 protein couple. Scale bars: 10 µm. *n* = 3/group (with > 30 cells/individual experiment). Controls and other protein couples are shown in Appendix Fig S4D and the original unsimplified photographs in Appendix Fig S4G.

Data information: Data are mean ± SEM. ****$P$ < 0.0001 vs. corresponding control (unpaired *t*-test).

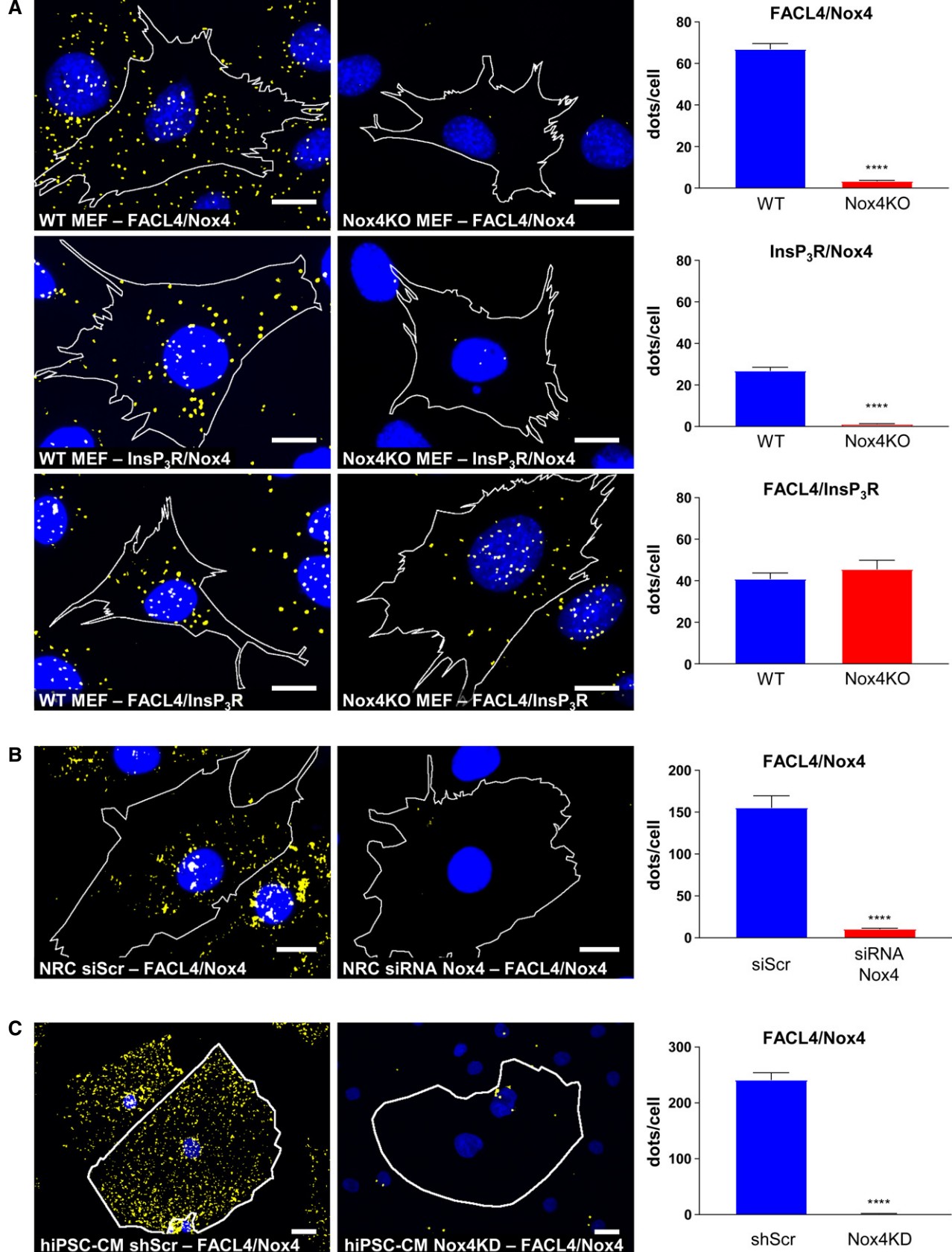

Figure 4.

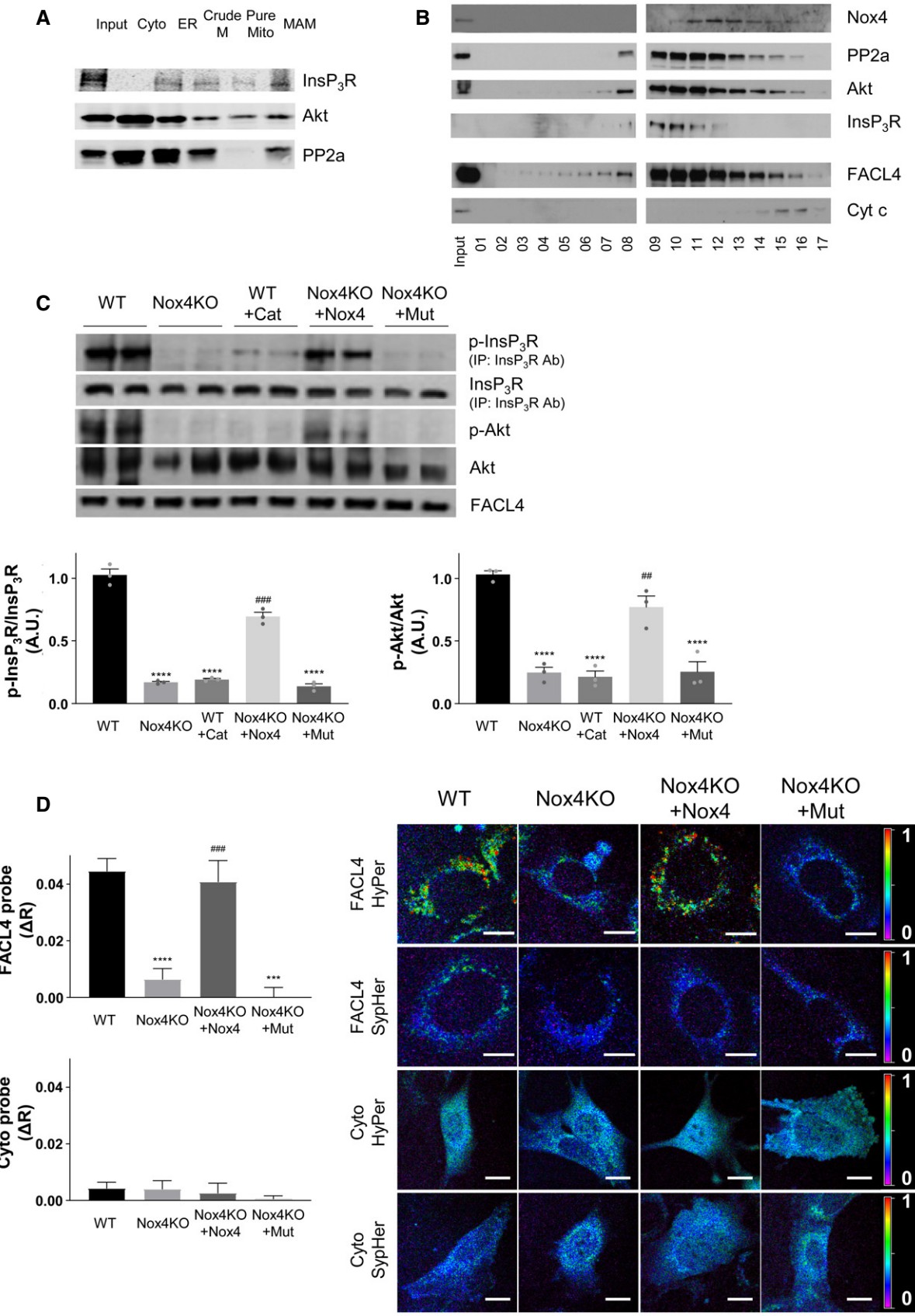

Figure 5.

**Figure 5. Regulation of Akt activation and InsP₃R phosphorylation by Nox4.**

A   Immunoblotting of subcellular fractions from serum-starved cardiomyoblasts for InsP₃R, Akt and PP2a, as in Fig 3A.

B   Fractionation of crude mitochondrial fractions (containing mitochondria and MAM) from WT MEFs on a sucrose gradient (5–60%) and immunoblotting for relevant proteins. Fraction size: 200 μl. Representative of 3 independent experiments.

C   Immunoblotting for phosphorylated and total Akt or InsP₃R protein in crude mitochondrial fractions from serum-starved WT and Nox4KO MEFs. FACL4 was used as a loading control for the MAM. InsP₃R was first immunoprecipitated, and then, the precipitate was immunoblotted for total InsP₃R and for the phosphorylated Akt substrate motif RXRXX(pS/T) (p-InsP₃R). Representative immunoblots are shown at the top and mean data at the bottom. Cat = PEG-catalase. Nox4KO MEFs were transfected either with active Nox4 or a Nox4$^{P437H}$ mutant (Mut). $n$ = 3/group.

D   ROS levels indexed in MEFs using $H_2O_2$ specific HyPer probes targeted to the MAM (FACL4 HyPer) or the cytosol (Cyto HyPer). Imaging was performed after serum starvation for 12 h. Representative photomicrographs of the fluorescence ratio are shown to the right and mean data for changes in fluorescence ratio presented to the left. The signal obtained using corresponding ROS-insensitive SypHer probes was used to correct for any pH-induced changes in fluorescence; the change in fluorescence ratio between HyPer and corresponding SypHer probe (ΔR) for each condition is reported. $n$ = 5 independent cell preparations/group, with at least 20 cells imaged/preparation. Scale bars: 10 μm.

Data information: Data are mean ± SEM. ***$P$ < 0.001; ****$P$ < 0.0001 among compared groups or vs. control. $^{###}P$ < 0.001; $^{##}P$ < 0.01 compared to Nox4KO (1-way ANOVA).

Source data are available online for this figure.

being attributable to PP2a based on the response to a PP2a inhibitor, okadaic acid, versus a combined PP1 and PP2a inhibitor, calyculin A (Fig EV4D). In Nox4 KO cells transfected with functional Nox4, the serine/threonine phosphatase activity was significantly decreased, whereas transfection with Nox4$^{P437H}$ had no effect (Fig EV4E). The incubation of serum-starved WT cells with PEG-catalase resulted in a significant increase in phosphatase activity (Fig EV4E).

The results so far are consistent with the notion that Nox4 may augment Akt activation via PP2a inhibition to increase InsP₃R phosphorylation and reduce calcium transfer to mitochondria during serum starvation, thereby protecting against cell death (Fig 6A). To test this sequence of events, we next undertook studies with different approaches to target Akt activation or the opening of the InsP₃R and then measure the effects on mitochondrial calcium and cell death. Treatment of serum-starved WT cells with a specific Akt inhibitor, Akti (Lindsley *et al*, 2005), significantly elevated basal mitochondrial calcium levels to a level similar to that observed in Nox4 KO cells, whereas Akti had no effect in Nox4 KO cells (Fig 6B). Similarly, the increase in mitochondrial calcium levels induced by histamine was markedly enhanced in Akti-treated WT cells as compared to WT control and approached the levels observed in Nox4 KO cells (Figs 6C and EV5A). The level of cell death was also similar in Akti-treated WT cells and Nox4 KO cells (Fig 6D). Akti had no significant effects in Nox4 KO cells. Treatment with the phosphatase inhibitor, okadaic acid, significantly increased both Akt and InsP₃R phosphorylation in Nox4 KO cells (Appendix Fig S6A). Okadaic acid also reduced the histamine-induced increase in mitochondrial calcium levels and cell death in serum-starved Nox4 KO cells (Fig EV5D and E). Treatment of cells with an InsP₃R inhibitor, xestospongin C (Gafni *et al*, 1997), almost completely inhibited histamine-induced calcium flux into the mitochondria, both in WT and Nox4 KO cells (Figs 6C and EV5A), and reduced cell death in Nox4 KO cells to levels similar to those observed in WT cells (Fig 6D). Xestospongin C had no effect on the phosphorylation level of Akt or InsP₃R (data not shown). Similar results were obtained in serum-starved cardiomyocytes depleted of Nox4 where xestospongin C significantly reduced the proportion of depolarized mitochondria and overall cell death (Fig EV5B and C). These results support the mechanistic sequence of events depicted in Fig 6A.

## RCD during ischemia–reperfusion injury of the heart is modulated by a Nox4/InsP₃R pathway

To investigate whether Nox4 mediates similar protective effects against RCD in a whole organ, we turned to a model of heart ischemia–reperfusion (I/R) injury, a setting in which mPT-mediated regulated cardiomyocyte necrosis is known to have a prominent role (Kung *et al*, 2011). We subjected hearts from Nox4-null mice and WT littermates to I/R and then measured the release of cardiac troponin I (cTnI) as a highly specific measure of cardiomyocyte death. We found that cTnI release was 7-fold higher after I/R injury of Nox4-null compared to WT hearts (Fig 6E). Quantification of the levels of phosphorylation of Akt and InsP₃R in cardiac tissue lysates containing the MAM fraction revealed that these were significantly lower in Nox4-null compared to WT hearts after I/R (Fig 6F). There was no difference in the phosphorylation levels of Akt and InsP₃R between non-ischemic WT and Nox4-null hearts (Appendix Fig S6B). To test whether InsP₃R was involved in this difference in cell death after I/R, we performed I/R after pre-incubation with xestospongin C. This showed that cTnI release from Nox4-null hearts was substantially reduced, whereas the InsP₃R blocker had minimal effect in WT hearts (Fig 6E). We also assessed the recovery of cardiac contractile function after I/R in these studies. Nox4-null hearts exhibited a lower recovery of contractile function than WT hearts, as assessed by the left ventricular rate-pressure product, left ventricular developed pressure, and other functional indices (Figs 6G and EV5F). There were no basal differences in contractile function between WT and Nox4-null hearts before ischemia. In the presence of xestospongin C, the extent of functional recovery was enhanced in both WT and Nox4-null hearts, with no difference between genotypes.

Taken together, these results indicate that Nox4-dependent inhibition of InsP₃R function during heart I/R and the consequent protection against cell death plays a crucial role in limiting cardiac I/R injury and facilitating recovery of contractile function.

## Discussion

Homeostatic programs that are activated in response to environmental stresses are of vital importance in determining cell viability. When such programs fail to restore homeostasis, the result may be

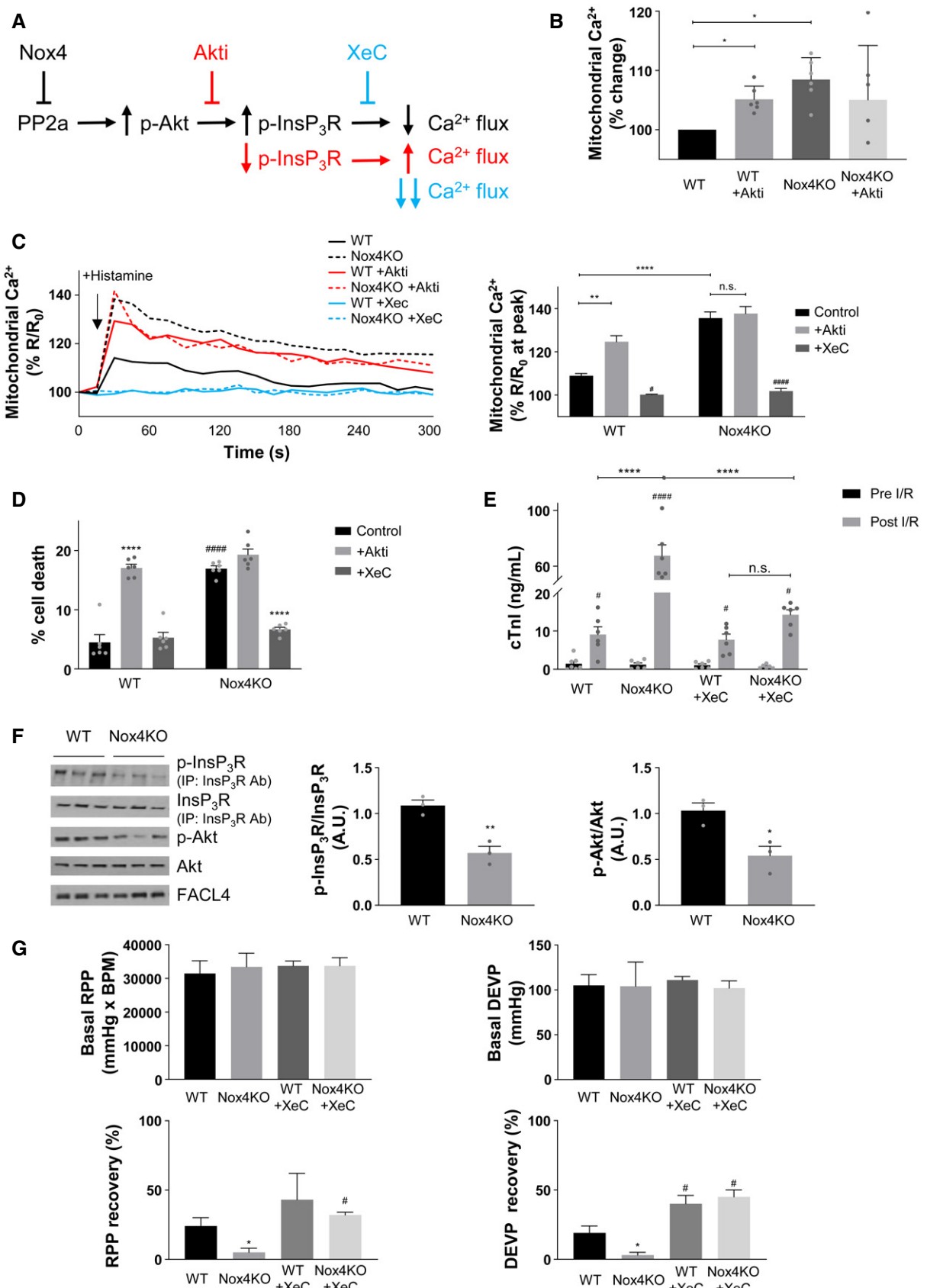

Figure 6.

**Figure 6. Functional impact of Nox4-dependent regulation of InsP₃R.**

A   Schematic of putative mechanism underlying Nox4-mediated regulation of calcium transfer via InsP₃R to mitochondria. Akti: Akt inhibitor; XeC: xestospongin C (InsP₃R blocker).

B   Basal mitochondrial calcium levels in serum-starved WT and Nox4KO MEFs in the presence or absence of Akti (1 μmol/l) normalized versus WT MEF control. n = 6/group with > 20 cells imaged per experiment.

C   Effect of Akti (1 μmol/l) or XeC (1 μmol/l) on histamine-induced changes in mitochondrial calcium levels in MEFs. The time course of changes in calcium levels is shown on the left and the mean data for peak mitochondrial calcium levels on the right. n = 3/group with > 40 cells imaged per experiment.

D   Quantification of cell death in serum-starved WT and Nox4KO MEFs in the absence or presence of Akti or XeC. n = 6/group.

E   Cardiac troponin I (cTnI) levels as a marker of cardiomyocyte necrosis in the perfusate of isolated WT and Nox4KO hearts subjected to ischemia–reperfusion (I/R). XeC was added at a final concentration of 2 μmol/l prior to ischemia. n = 6–7/group.

F   Immunoblotting for the phosphorylation levels of Akt and InsP₃R in crude mitochondrial fractions from WT and Nox4KO hearts after I/R. InsP₃R was first immunoprecipitated, and then, the precipitate was immunoblotted for total InsP₃R and for the phosphorylated Akt substrate motif RXRXX(pS/T) (p-InsP₃R). Mean data shown to the right. n = 3/group.

G   Cardiac left ventricular contractile function in isolated Langendorff-perfused WT and Nox4KO hearts at baseline (Basal) and then after I/R. Hearts were treated with XeC (1 μmol/l) or vehicle control for 20 min prior to ischemia. RPP, heart rate × left ventricular pressure product; DEVP, left ventricular developed pressure. n = 6–7/group.

Data information: Data are mean ± SEM. *$P < 0.05$, **$P < 0.01$; ****$P < 0.0001$ among compared groups or vs. control. #$P < 0.05$; ####$P < 0.0001$ among different groups within the same treatment or among different treatments within the same genotype. (B,G): 1-way ANOVA; (C-E): 2-way ANOVA; (F): unpaired t-test.

the triggering of RCD, a process in which mitochondria play a central role (Galluzzi et al, 2014). Mitochondrial calcium levels are a key determinant of RCD since the mitochondrial calcium machinery responds to and decodes diverse stress signals and in turn has a major impact on the mPTP (Giorgi et al, 2018). Here, we identify Nox4-dependent inhibition of calcium release via InsP₃R to mitochondria as a potent pro-survival pathway, involving spatially confined signaling at the MAM to augment Akt-dependent phosphorylation of InsP₃R (Fig 7). This Nox4-dependent homeostatic pathway is highly effective in preventing mPT-dependent RCD in cells and mitigating injury and contractile impairment after ischemia–reperfusion of the heart.

The MAM is well recognized as a key hub that regulates diverse cellular functions by mediating cross-talk between the ER and mitochondria, for example, through the transfer of calcium (Filadi et al, 2017; Csordas et al, 2018; Marchi et al, 2018). Proteins that are enriched at the MAM play important roles in the inter-organelle communication and regulation between ER and mitochondria, and the overall impact of this subcellular domain on cellular metabolism and viability. InsP₃R, VDAC, and the MCU are all concentrated at the MAM where calcium microdomains of ~10-fold higher concentration than in the bulk cytosol are generated. InsP₃R are crucial for physiological calcium transfer from ER to mitochondria during normal homeostasis but enhanced calcium transfer also plays a major role in many types of RCD under stress conditions (Roest et al, 2017). The regulation of the calcium release activity of InsP₃R is therefore an important determinant of cell fate. Previous studies reported an ER localization of Nox4 (Ambasta et al, 2004; Anilkumar et al, 2008; Santos et al, 2016) without assessing whether it may be enriched in specific domains. We found with the use of several complementary approaches that Nox4 is highly enriched at the MAM in close proximity to other key players in the homeostatic pathway described herein. Interestingly, it was previously reported that Nox4 interacts with the ER chaperone calnexin (Prior et al, 2016) which is itself enriched at the MAM. These findings do not exclude other subcellular locations for these proteins under different conditions, and it is indeed well recognized that the MAM itself is a highly dynamic structure.

The MAM localization of Nox4 positions it to exert optimal spatially confined regulatory control over calcium transfer to mitochondria via InsP₃R. We found that the mechanism underlying this regulation was through the modulation of Akt phosphorylation which in turn phosphorylates InsP₃R at the MAM to control their calcium release activity. Previous studies showed that Akt phosphorylation of type 3 InsP₃R, which are concentrated at the MAM, inhibits calcium transfer through this channel (Marchi et al, 2012). Of note, type 3 InsP₃R are reported to be insensitive to modulation by protein kinase A (Soulsby & Wojcikiewicz, 2007). The modulation of Akt activation by Nox4 is likely to involve the oxidative inactivation of the serine/threonine phosphatase PP2a, which otherwise dephosphorylates Akt, analogous to redox signaling mechanisms that modulate the phosphorylation of other proteins (Foley et al, 2007; Santos et al, 2016). The pro-survival effects of Nox4 contrast to the detrimental effects of oxidative stress described in other settings. For example, high levels of ROS can induce cytosolic and mitochondrial calcium overload and are also a potent stimulus for the mPT (Izzo et al, 2016). This apparent contradiction is explained by the spatially confined redox signaling mediated by Nox4 at the MAM. Notably, changes in Nox4-mediated ROS detected at the MAM are not accompanied by changes in ROS levels in the bulk cytosol or the mitochondrial matrix and it is likely that the signaling triggered by Nox4 involves ROS production in highly limited microdomains. The mechanism underlying the stress-induced increase in Nox4 levels was not addressed here but previous work has identified that the transcription factor ATF4, which is induced by numerous environmental stresses, upregulates Nox4 (Santos et al, 2016).

The inhibition of mPT-dependent RCD by Nox4 was also observed in MEFs, suggesting that it may be relevant to many cell types. However, it is likely to be especially important in post-mitotic somatic cells with limited regenerative capacity such as cardiomyocytes and neurons. We found that Nox4 not only inhibited RCD evoked by serum starvation in cultured cardiomyocytes but also had a major impact on cardiomyocyte death in a model of whole heart I/R injury. As such, the hearts of Nox4-null mice had substantially higher release of the cardiomyocyte-specific protein troponin I (a specific marker of cardiomyocyte cell death) after I/R than WT hearts. This difference between Nox4-null and WT was abrogated by an inhibitor of InsP₃R channels, xestospongin C, indicating that the mechanism underlying the protective effects of Nox4 involved InsP₃R channels—a conclusion that was also supported by Nox4-

**WT**

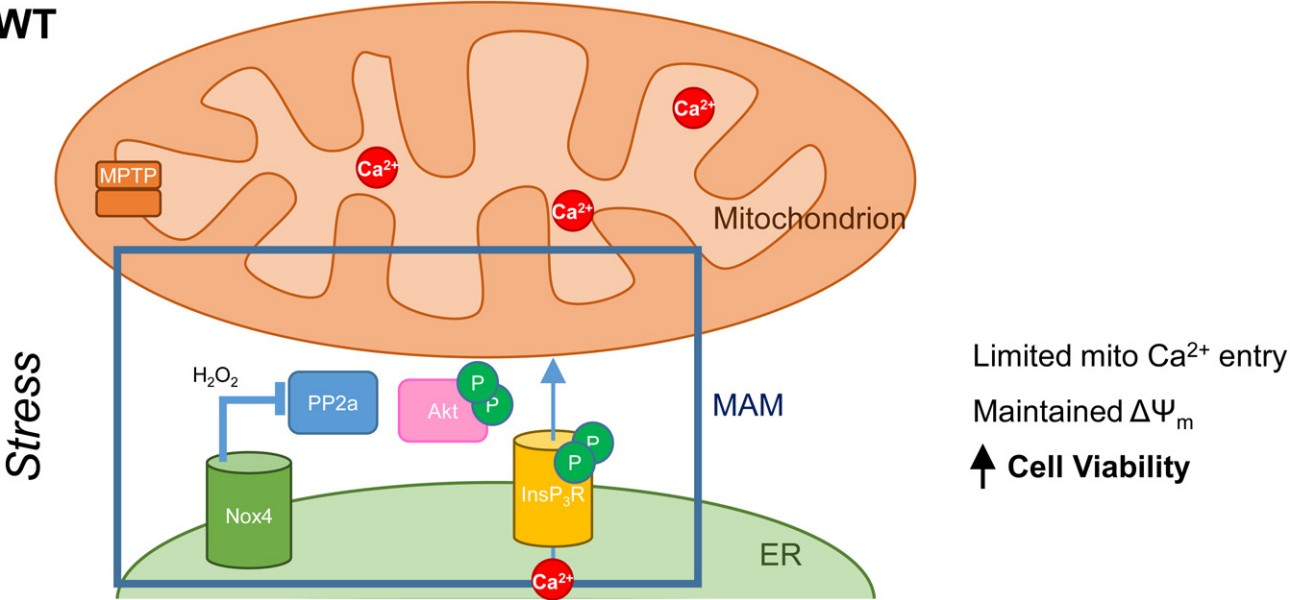

Limited mito Ca²⁺ entry

Maintained ΔΨₘ

↑ **Cell Viability**

**Nox4KO**

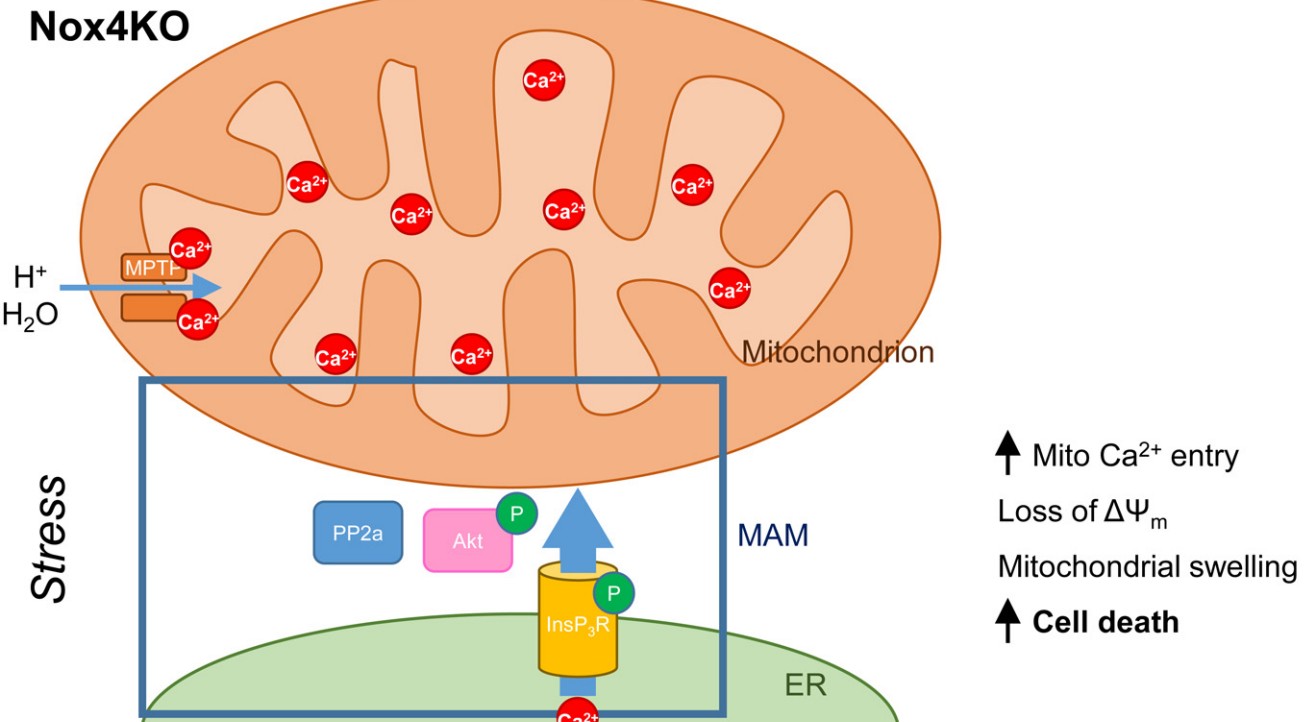

↑ Mito Ca²⁺ entry

Loss of ΔΨₘ

Mitochondrial swelling

↑ **Cell death**

**Figure 7.**

**Figure 7.  Schematic depicting the regulation by Nox4 of calcium transfer into mitochondria during cell stress.**

During cellular stress, Nox4 at the MAM augments the level of Akt activation secondary to redox inhibition of PP2a. Activated Akt in turn phosphorylates InsP$_3$R and leads to an inhibition of calcium flux from ER to mitochondria. Mitochondrial calcium is maintained at a low level, the mitochondrial membrane potential ($\Delta\Psi_m$) is preserved, and cells remain viable. When Nox4 is deficient (bottom panel), high PP2a activity dephosphorylates Akt and leads to a reduced level of InsP$_3$R phosphorylation. This removes the restraint on calcium transfer from ER to mitochondria resulting in increased mitochondrial calcium levels, triggering of the mPT, loss of $\Delta\Psi_m$, and eventual cell death.

dependent changes in the phosphorylation of Akt and InsP$_3$R in heart tissue. Furthermore, the impairment of cardiac contractile function that is typically observed after I/R injury was worse in Nox4-null hearts but was markedly improved by the InsP$_3$R blocker. These findings may have relevance to myocardial infarction, a condition that results from the sudden occlusion of a coronary artery supplying the heart muscle and is typically treated by the restoration of coronary blood flow after a few hours. Myocardial infarction causes major human mortality and morbidity and is frequently complicated by a significant impairment of cardiac contractile function even after restoration of coronary blood flow. I/R injury as assessed in the current study simulates myocardial infarction, and our results suggest not only that a Nox4-Akt-InsP$_3$R pathway may be important in mitigating injury but also that targeting the pathway (e.g., with an InsP$_3$R blocker) could have therapeutic potential. Further studies in humans would however be required to address this possibility.

In conclusion, this study identifies a Nox4-dependent stress pathway that protects against RCD through spatially delimited redox signaling at the MAM to inhibit the calcium release activity of InsP$_3$R. The findings provide a molecular insight into the regulation of the pathway in somatic cells and may be especially relevant for post-mitotic cells like cardiomyocytes that are prone to frequent death-promoting stresses (such as myocardial ischemia and infarction) in humans.

# Materials and Methods

All chemicals were purchased from Merck/Sigma-Aldrich (Gillingham, UK) unless otherwise indicated and were of analytical or higher purity grade.

### Cell culture and gene expression manipulation

Primary cultures of cardiomyocytes were prepared from neonatal rats (Zhang *et al*, 2010). Rat H9c2 cardiomyoblasts were purchased from ATCC via LGC Standards (Teddington, UK). Nox4KO and WT MEFs were described previously (Santos *et al*, 2016). Human-induced pluripotent stem cell-derived cardiomyocytes (hiPSC-CM) were prepared as described (Borchert *et al*, 2017). Cardiomyocytes were cultured in a medium with varying amounts of serum, from 0% (serum-free medium) to 15% (complete medium: 10% FCS, fetal calf serum + 5% horse serum). H9c2 cells and MEFs were grown either in serum-free medium (0%) or complete medium (10% FCS). hiPSC-CM were grown in RPMI 1640 culture medium supplemented with 2 mM L-glutamine and 1× B27 with insulin (Thermo Fisher Scientific, Loughborough, UK), without serum. Serum starvation (in combination with other stimuli or gene expression manipulation) was performed for up to 48 h unless otherwise indicated. Drugs were added before harvesting of cells or

measurements as follows: Akti for 3 h at a final concentration of 1 μmol/l; xestospongin C (Abcam, Cambridge, UK) for 20 min at a final concentration of 1 μmol/l; PEG-catalase for 48 h at a final concentration of 500 U/ml; cyclosporin A (CsA) at a final concentration of 1 μmol/l for 24 h; bongkrekic acid at a final concentration of 5 μmol/l) for 24 h; okadaic acid at a final concentration of 0.5 μmol/l for 1 h; or Mito-TEMPO (2-(2,2,6,6-Tetramethylpiperidin-1-oxyl-4-ylamino)-2-oxoethyl)triphenyl phosphonium chloride) at a final concentration of 20 μmol/l for 24 h.

Cells were transfected with plasmids or siRNA using TurboFect (Thermo Fisher Scientific, for MEFs) or transfectin (BioRad, Watford, UK, for cardiomyocytes). Plasmids used for the study were as follows: pCDNA3.1-Nox4 (Anilkumar *et al*, 2008); pcDNA3.1-Nox4$^{P437H}$ (Santos *et al*, 2016); pcDNA-4mtD3cpv (Addgene, Cambridge, MA, USA, #36324); pcDNA-D1ER (Addgene #36325); pC1-HyPer-cyto, pC1-SypHer-cyto, pC1-HyPer-mito and pC1-SypHer-mito (Belousov *et al*, 2006); pcDNA-FACL4-HyPer and pcDNA-FACL4-SypHer, generated by PCR to add the ROS probes in frame at the 3′ end of FACL4. Short interfering RNAs (siRNAs) for rat Nox4 were purchased from Qiagen (Manchester, UK—siRNA#1) or Ambion (via Thermo Fisher Scientific—siRNA#2). Sequences used were as follows: #1 sense: 5′-CCAUUAUCUCAGUAAUCAATT, antisense: 5′-UUGAUUACUGAGAUAAUGGTG; #2: sense: 5′-CCGUUUGCAUC GAUACUAATT, antisense: 5′-UUAGUAUCGAUGCAAACGGAG.

Adenoviruses expressing a short hairpin sequence targeted against Nox4 (Ad.shNox4) or a short hairpin sequence targeted against GFP as control (Ad.Ctl; Peterson *et al*, 2009) were used to infect cardiomyocytes at a multiplicity of infection (MOI) of 20. Cells were used 48 h later. Lentiviruses expressing a short hairpin sequence targeted against human Nox4 or a scramble short hairpin sequence were purchased from OriGene (Rockville, MD, USA; Cat. no #TL302911V) and used to infect hiPSC-CM at an MOI of 10. Cells were used 72 h later.

### Real-time RT–PCR

Total RNA was isolated using the ReliaPrep RNA Miniprep (Promega, Southampton, UK), according to the manufacturer's protocol. mRNA expression levels were quantified using the following primers: rat Nox4 F: 5′-AGCTCATTTCCCACAGACCT, R: 5′-TCCGGATGCATCGGTAAAGT; rat β-actin F: 5′-CCCGCGAGTACA ACCTTCT, R: 5′-CGTCATCCATGGCGAACT. Quantitative real-time PCR was performed with a SybrGreen mix (PCR Biosystems, London, UK) on a StepOne Plus Real-Time PCR thermal cycler (Applied Biosystems, Warrington, UK), using the ΔΔCt method to calculate relative fold change.

### Subcellular fractionation

Subcellular fractions (cytosol, ER, crude mitochondria, pure mitochondria, MAM) of cells and tissues were isolated as described

previously (Wieckowski *et al*, 2009). Briefly, H9c2 cells were detached from $20 \times$ T175 flasks using trypsin, washed with PBS, and resuspended in cell-isolation buffer (225 mmol/l mannitol, 75 mmol/l sucrose, 0.1 mmol/l EGTA and 30 mmol/l Tris–HCl pH 7.4). Tissues were homogenized to a final of concentration 0.25 g/ml with the same isolation buffer but also containing 0.5% BSA. Cells and tissues were homogenized with a Teflon pestle at 4,000 rpm. Unbroken cells and nuclei were pelleted at 600 *g* for 5 min at 4°C. Supernatant was further spun at 7,000 *g* for 10 min at 4°C to pellet crude mitochondria, which were then used for sucrose gradient or immunoblotting analysis, after resuspending them in isolation buffer (100 mmol/l Tris pH 7.2, 20 mmol/l $MgCl_2$, 15 mmol/l KCl, 0.1 mmol/l EDTA, 0.1 mmol/l EGTA, containing protease and phosphatase inhibitor cocktails). For further fractionation, the supernatant (obtained after centrifugation at 7,000 *g* for 10 min at 4°C) was ultra-centrifuged at 100,000 *g* for 45 min at 4°C to obtain ER (pellet) and cytosolic fraction (supernatant). Crude mitochondria were gently washed twice with mitochondrial buffer (225 mmol/l mannitol, 75 mmol/l sucrose, and 30 mmol/l Tris pH 7.4 [with 0.5% BSA in the case of tissue]). After washing, samples were ultra-centrifuged on a Percoll gradient at 95,000 *g* for 30 min at 4°C. The pure mitochondria appeared as a pellet in the very bottom of the tube, and the fraction containing MAM was a diffuse white band located above the mitochondria. The pure mitochondria were spun at 6,300 *g* for 10 min at 4°C to remove MAM contamination, while the MAM fraction was concentrated by ultra-centrifuging at 100,000 *g* for 1 h at 4°C. Finally, ER, pure mitochondria and MAM fractions were all diluted in resuspension buffer (250 mmol/l mannitol, 5 mmol/l HEPES pH 7.4, and 0.5 mmol/l EGTA, with protease and phosphatase inhibitor cocktails).

## Sucrose gradient fractionation

Mouse embryonic fibroblasts grown in 10 mm Petri dishes were scraped (4 dishes/sample), transferred into tubes, and centrifuged at 3,000 *g* at 4°C for 5 min. The cell pellet was resuspended in 250 µl lysis buffer (50 mmol/l Tris pH 8.0, 150 mmol/l NaCl, 0.1% SDS, and 1% Nonidet NP40, containing protease cocktail). Cell lysates were laid at the top of the sucrose gradient (5%, 10%, 20%, 40%, and 60%, top to bottom), which was prepared in 50 mmol/l HEPES buffer pH 7.5, containing 100 mmol/l KCl, 2 mmol/l $MgCl_2$, 1 mmol/l EGTA, and 1 mmol/l EDTA. Samples were centrifuged at 35,000 *g* (4°C, 18 h). Fractions F1-F17 (200 µl each) were collected from the base of the column and used for immunoblotting.

## Immunoblotting

Cells were harvested in culture medium, centrifuged at 3,000 *g* for 5 min, and pellets resuspended in lysis buffer (50 mmol/l Tris pH 8.0, 150 mmol/l NaCl, 0.1% SDS, and 1% Nonidet NP40, containing protease and phosphatase inhibitor cocktails). For HMGB1 release, cell media were collected after 48 h serum starvation, snap-frozen in liquid nitrogen, and freeze-dried using an Alpha 1-2 LD plus freeze-drier (Martin Christ Gefriertrocknungsanlagen GmbH, Osterode am Harz, Germany). Samples were then resuspended in 100 µl distilled water. To detect phosphorylation of $InsP_3R$ at its Akt substrate motif RXRXX(S/T), $InsP_3R$ was first immunoprecipitated from crude mitochondrial fractions (containing MAM) using A/G

Sepharose beads (Santa Cruz) as described previously (Khan *et al*, 2006). Mouse IgG was used as a control. The immunoprecipitate was then immunoblotted using an antibody against the type 3 $InsP_3R$ to detect total $InsP_3R$ and in parallel against the phospho-Akt substrate motif to detect phosphorylated $InsP_3R$. Protein concentration was determined with the Pierce BCA Protein Assay Kit (Thermo Fisher Scientific). Equal amounts of lysates, sucrose gradient fractions, subcellular fractions, or concentrated media were loaded onto SDS–polyacrylamide gels (Bolt™ Bis-Tris Plus Gels, Thermo Fisher Scientific), run at 150 V for 30–60 min, and then transferred onto nitrocellulose membranes. Membranes were blocked with Tris-buffered saline and 0.05% Tween 20 (TBST) containing 5% non-fat milk, then incubated overnight with primary antibodies at 4°C. IRDye 680RD- or IRDye 800CW-conjugated secondary antibodies (Li-Cor, Cambridge, UK) were used and signals detected with an Odyssey CLX Blot imager (Li-Cor). Densitometric analysis was performed using Image Studio 5.2 software (Li-Cor).

Antibodies used were (in alphabetical order) as follows: AIF (rabbit polyclonal, 1:1,000, Cell Signaling Technology, Hitchin, UK); phospho-AKT (S473, rabbit monoclonal, 1:1,000, Cell Signaling Technology); total AKT (rabbit polyclonal, 1:1,000, Cell Signaling Technology); phospho-Akt substrate motif RXRXX(S/T) (23C8D2, rabbit monoclonal #10001; 1:1,000, Cell Signaling Technology); calnexin (rabbit polyclonal, 1:1,000, Abcam); caspase 3 (rabbit monoclonal, 1:1,000, Cell Signaling Technology); caspase 8 (rabbit monoclonal, 1:1,000, Cell Signaling Technology); cytochrome c (rabbit monoclonal, 1:1,000, Cell Signaling Technology); FACL4 (rabbit polyclonal, 1:1,000, Abcam); HMGB1 (rabbit polyclonal, 1:1,000, Abcam); $InsP_3R$ type 3 (mouse monoclonal, 1:1,000, Becton Dickinson & Co Biosciences, Wokingham, UK); Nox4 (rabbit polyclonal, 1:3,000; Anilkumar *et al*, 2008); PAR (mouse monoclonal, 1:1,000, Enzo Life Sciences, Exeter, UK); PP2A (Subunit c, rabbit polyclonal, 1:1,000, Cell Signaling Technology); RIP1 (mouse monoclonal, 1:1,000, R&D Systems, Abingdon, UK); RIP3 (mouse monoclonal, 1:1,000, R&D Systems); Sigma1R (rabbit monoclonal, 1:1,000, Cell Signaling Technology); tubulin (mouse monoclonal, 1:10,000, Sigma-Aldrich); and VDAC (rabbit polyclonal, 1:1,000, Cell Signaling Technology).

## Cell viability assays

Cell viability was visualized through double staining with Hoechst 33342 (Thermo Fisher Scientific), to identify all nuclei, and 7-AAD (Thermo Fisher Scientific), which is excluded by viable cells, using fluorescence microscopy. Cells were incubated with 0.8 µg/ml Hoechst 333342 and 1 µg/ml 7-AAD in medium for 15 min. Images were taken at 37°C/5% $CO_2$ on an inverted Nikon Ti-E microscope equipped with a Yokogawa CSU-X1 spinning-disk confocal unit (Nikon Instruments, Kingston Upon Thames, UK). Dyes were excited at 405 nm and visualized with emission at 460/50 nm (Hoechst 33342), or excited at 561 nm and visualized with emission at 645/75 nm (7-AAD). Red positive and blue nuclei were quantified using ImageJ software v1.8.0-112 (NIH, Bethesda, MD, USA) to calculate the percentage of cell death. At least 300 cells/condition were counted in three independent experiments.

Lactate dehydrogenase (LDH) release into the medium was quantified with a CytoTox-ONE Homogenous Membrane Integrity Assay kit (Promega) following the manufacturer's instructions.

## Mitochondrial membrane depolarization

Mitochondrial membrane potential was measured using tetramethylrhodamine ethyl ester (TMRE) and flow cytometry as described previously (Crowley *et al*, 2016). Briefly, cells (approx. 500,000/well) were incubated with 400 nmol/l TMRE in medium for 30 min. Cells were collected and resuspended into 400 µl PBS + 0.2% BSA. A BD Accuri C6 flow cytometer (Becton Dickinson & Co Biosciences) was used to measure $\Delta\Psi_m$, using the 488 nm excitation laser and 585/40 nm (FL2) emission detector. Unstained cells were run as negative controls. 100,000 cells were counted for each sample. The percentage of cells with depolarized mitochondria was calculated as the percentage of cells that lost the dye (were negative) after treatment. Analysis was done with FlowJo V10 software (FlowJo LLC, Ashland, OR, USA).

## Intracellular calcium measurements

Mitochondrial calcium was measured as described previously (Palmer & Tsien, 2006). Cells were seeded on 35 mm imaging dishes (Ibidi, Martinsried, Germany), transfected with a cameleon sensor targeted to the mitochondria (pcDNA-4mtD3cpv), and, in case of cardiomyocytes, co-transfected with siRNAs. Cells were incubated in phenol red-free medium supplemented with 2 mmol/l L-glutamine and antibiotics in the absence or presence of serum for 24 h before imaging. Images were taken at 37°C/5% $CO_2$ on an inverted Nikon Ti-E A1R confocal microscope using a Nikon 40× Plan Apo VC NA 1.40 water immersion objective (Nikon Instruments). Fluorescence emission was monitored at 482/35 nm (CFP) and 540/30 nm (YFP) following excitation at 445 nm, and the ratio (R) of YFP and CFP fluorescence intensity was quantified with NIS Elements v.5.0 software (Nikon Instruments), after background-subtraction and threshold setting. The resulting images were displayed in pseudocolor. $InsP_3R$-mediated calcium release from ER stores was triggered by the addition of 100 µmol/l histamine or 100 µmol/l ATP. Caffeine (to stimulate calcium release via ryanodine receptors) was added at a final concentration of 10 mmol/l immediately before recording. In this set of experiments, mitochondrial calcium levels were calculated as relative YFP/CFP ratio compared to baseline ratio at the start of the measurement (% $R/R_0$). In experiments with TMRE, this was loaded for 15 min at a concentration of 20 nmol/l before washing with phenol red-free medium and imaging. The TMRE image was briefly acquired to distinguish between positive and negative cells, and then, imaging for calcium was performed as described above.

For ER calcium measurements, we used a cameleon sensor targeted to the ER (pcDNA-4D1ER; Palmer *et al*, 2004). To avoid ER reloading of calcium from culture medium, cells were incubated in HBSS (Thermo Fisher Scientific) supplemented with 2 mmol/l L-glutamine and antibiotics in the absence of serum before imaging and detection of fluorescence emission as described above. ER calcium levels were calculated as relative YFP/CFP ratio compared to baseline ratio at the start of the measurement (% $R/R_0$).

Cytosolic calcium measurements were performed using Fluo4-AM (De Vos *et al*, 2012). Cells were loaded with 2 µmol/l Fluo4-AM (Thermo Fisher Scientific) in solution containing 0.02% Pluronic F27 (Thermo Fisher Scientific) for 30 min at 37°C. Fluo4 (green) fluorescence was recorded (at 1s intervals) at 37°C using a Nikon Eclipse Ti-E microscope equipped with an Andor Neo sCMOS camera, Chroma filter sets, a Nikon 40× Plan Fluor oil DIC NA 1.30 W.D. 0.2 mm objective (Nikon Instruments), and a MSC200 fast perfusion system (Bio-Logic Science Instruments, Seyssinet-Pariset, France). Fluo4 was excited at 488 nm and emission visualized at 525/40 nm. Cells were kept under constant perfusion (0.5 ml/min) with external solution (145 mmol/l NaCl, 2 mmol/l KCl, 5 mmol/l $NaHCO_3$, 1 mmol/l $MgCl_2$, 2.5 mmol/l $CaCl_2$, 10 mmol/l glucose, 10 mmol/l Na-HEPES pH 7.25, 0.02% Pluronic F27). Cytosolic calcium levels were calculated as relative Fluo4-AM fluorescence compared to baseline fluorescence at the start of the measurement (% $F/F_0$).

## Transmission electron microscopy (TEM)

Cardiomyocytes grown on coverslips were fixed overnight with 2.5% (v/v) glutaraldehyde in 0.1 mol/l phosphate buffer (pH 7.3) at 4°C. Cardiac muscle tissue samples were perfusion-fixed with 2% (v/v) glutaraldehyde, 2% (w/v) paraformaldehyde, 2.5% (w/v) polyvinylpyrrolidone, 0.1% (w/v) sodium nitrite in 0.1 mol/l PIPES buffer. After initial fixation, samples were rinsed several times in 0.1 mol/l PIPES buffer and post-fixed with 1% osmium tetroxide in 0.1 mol/l phosphate buffer for 1.5 h at 4°C. Samples were then washed, dehydrated in a graded series of ethanol, and equilibrated with propylene oxide. Cells/tissue were then infiltrated with epoxy resin (TAAB Laboratories Equipment Ltd, Aldermaston, UK) and cured at 70°C for 24 h. Ultrathin sections (70–90 nm) were prepared using an Ultracut E ultramicrotome (Reichert-Jung, Leica Microsystems Ltd, Milton Keynes, UK), mounted on 150 mesh copper grids, contrasted using uranyl acetate and lead citrate, and examined on a Tecnai 12 transmission electron microscope (FEI, Thermo Fisher Scientific) operated at 120 kV. Images were acquired with an AMT 16000M camera (Advanced Microscopy Techniques, Woburn, MA, USA).

For TEM immunogold labeling, tissue samples were perfused-fixed with 2% (w/v) paraformaldehyde, 0.2% (v/v) glutaraldehyde, 2.5% (w/v) PVP, 0.1% (w/v) sodium nitrite in 0.1 mol/l PIPES. Cell samples were fixed with 4% (w/v) paraformaldehyde, 0.1% (v/v) glutaraldehyde in 0.1 mol/l phosphate buffer (pH 7.4) for 4 h at room temperature and spun down on 20% gelatin. All samples were cut into smaller pieces and cryoprotected by incubating in 2.3 mol/l sucrose overnight at 4°C. Samples mounted on aluminum pins were then cryofixed by plunging into liquid nitrogen and stored in liquid nitrogen prior to cryosectioning. Ultrathin sections (70–90 nm thick) were cut using a Leica EM FC6 cryo-ultramicrotome (Leica Microsystems) and mounted on pioloform film-supported nickel grids (Tokuyasu, 1980). Sections were immuno-labeled with FACL4 (rabbit polyclonal, 1:1,000, Abcam), rodent Nox4 (rabbit polyclonal, 1:1,000; Anilkumar *et al*, 2008), or human Nox4 (rabbit monoclonal, 1:1,000; Meitzler *et al*, 2017) followed by 12 nm-colloidal gold secondary antibody (Jackson ImmunoResearch, Ely, UK) at 1:40. Sections were examined using either a Tecnai 12 TEM (FEI, Thermo Fisher Scientific) operated at 120 kV fitted with an AMT 16000M digital camera (Advanced Microscopy Techniques), or a JEM1400-Plus transmission electron microscope (JEOL, Welwyn Garden City, UK), fitted with a Ruby digital camera (JEOL).

## Confocal microscopy

Cells were grown on 14 mm round coverslips and fixed in 3% formaldehyde in PBS pH 7.4 containing 0.025% glutaraldehyde. Cells were permeabilized with 0.1% Triton and incubated with primary antibodies. Antibodies used were as follows: FACL4 (mouse monoclonal, 1:500, Santa Cruz Biotechnology Inc.); InsP$_3$R type 3 (mouse monoclonal, 1:500, Becton Dickinson Biosciences); rodent Nox4 (rabbit polyclonal, 1:500; Anilkumar *et al*, 2008). Secondary antibodies were as follows: goat anti-mouse Alexa 568 and goat anti-rabbit Alexa 488 (both 1:500, Thermo Fisher). Cells were mounted in Mowiol and imaging was performed on an inverted Nikon Ti-E microscope equipped with a Yokogawa CSU-X1 spinning-disk confocal unit, an Andor DU-897 camera, a Sutter filter wheel, and Nikon 100 × 1.49 NA objective (Nikon Instruments).

## Proximity ligation assay

Proximity ligation assays were performed using Duolink technology (De Vos *et al*, 2012). Cells were fixed in 4% paraformaldehyde in PBS at room temperature for 20 min, permeabilized with 0.2% Triton X100 in PBS at room temperature for 3 min, and probed with primary antibodies (alone for controls, or in pairs). Signals were developed using the Duolink® *In Situ* Green kit and quantified (dots/cell) using ImageJ software v1.8.0-112 (NIH). Cells were co-stained with DAPI (1:200) for nuclear detection and with Phalloidin CruzFluor 555 conjugate (1:500, Santa Cruz Biotechnology Inc., Heidelberg, Germany) for the cytoskeleton. Images were acquired using a Leica TCS-SP5 confocal microscope with a 63× HCX PL APO lambda blue CS 1.4 oil UV objective (Leica Microsystems). Images were collected using single excitation for each wavelength separately (405 nm UV diode and a 422–470 nm emission band pass for DAPI; 488 nm Argon Laser line and a 500–545 nm emission band pass for the Duolink Green reagent; 561 nm DPSS Laser line and a 585–690 nm emission band pass for Phalloidin CruzFluor 555 conjugate). Ten to fifteen image sections of selected areas were acquired in each experiment and analyzed (dots/cell) using ImageJ software v1.8.0-112 (NIH).

Antibodies used were (in alphabetical order) as follows: FACL4 (mouse monoclonal, 1:500, Santa Cruz Biotechnology Inc.); FACL4 (rabbit polyclonal, 1:500, Abcam); InsP$_3$R type 3 (mouse monoclonal, 1:500, Becton Dickinson & Co Biosciences); LAMP1 (mouse monoclonal, 1:500, Santa Cruz Biotechnology Inc.); rodent Nox4 (rabbit polyclonal, 1:500; Anilkumar *et al*, 2008); human Nox4 (rabbit monoclonal, 1:500; Meitzler *et al*, 2017).

## Serine/threonine phosphatase activity

The phosphatase activity in crude mitochondria was measured using a Malachite Green assay (Millipore, Watford, UK) based on the hydrolysis of a phospho-threonine peptide (Lys-Arg-phosphoThr-Iso-Arg-Arg). Crude mitochondria from MEFs (5–10 μg) were incubated with phosphopeptide substrate (0.1 mmol/l) for 20 min at 37°C; then, 100 μl of Malachite Green reagent was added and the reaction followed at 620 nm using a Nanodrop spectrophotometer (Thermo Fisher Scientific). Some experiments were performed in the presence of okadaic acid (OA, 10 nmol/l), which does not inhibit PP1 at this concentration (Ishihara *et al*, 1989),

calyculin A (60 nmol/l), which inhibits both PP1 and PP2a (Ishihara *et al*, 1989), or H$_2$O$_2$ (500 μmol/l for 15 min). PP2a activity was calculated as the difference between control and OA-treated samples, while PP1 + PP2a activity was the difference between control and calyculin A-treated samples after normalization for protein content and blank subtraction.

## H$_2$O$_2$ measurements

Intracellular ROS were visualized using HyPer and SypHer probes (Belousov *et al*, 2006). The redox-sensitive Cys199 residue in HyPer proteins is mutated to serine in SypHer probes, which were used as negative controls to exclude changes in pH. WT and KO MEFs were seeded on 35 mm imaging dishes (Ibidi), transfected with HyPer-cyto, FACL4-HyPer, or mitochondrial-targeted HyPer and kept in phenol red-free medium supplemented with 2 mmol/l L-glutamine and antibiotics for 12 h before imaging. Images were taken at 37°C/5% CO$_2$ on an inverted Nikon Ti-E microscope equipped with a Yokogawa CSU-X1 spinning-disk confocal unit, an Andor DU-897 camera, a Sutter filter wheel, and Nikon 40× Plan Apo VC NA 1.40 water immersion objective (Nikon Instruments). Fluorescence emission was monitored at 535/40 nm following excitation at 405 and 488 nm, and the ratio (R) of fluorescence intensity was quantified. Extracellular H$_2$O$_2$ (200 μmol/l) was added as a positive control. NIS Elements v.5.0 software (Nikon Instruments) was used for image analysis. Images were background-subtracted and thresholded. Changes in fluorescence ratio between HyPer and the corresponding SypHer probes (ΔR) between the indicated genotypes and treatments were quantified. The resulting images were displayed in pseudocolor.

H$_2$O$_2$ produced by cells after overexpression of Nox4 was measured using an Amplex Red Hydrogen Peroxide/Peroxidase Assay Kit (Thermo Fisher Scientific) according to the manufacturer's instructions.

## Animal studies

All procedures were performed in compliance with the UK Home Office "Guidance on the Operation of the Animals" (Scientific Procedures) Act, 1986 and institutional guidelines. Nox4 KO mice (on a C57BL6 background) were described previously (Zhang *et al*, 2010). Heart ischemia/reperfusion (I/R) injury was assessed in hearts of Nox4 KO animals and matched WT littermates. Hearts were perfused with a modified KHB buffer on a Langendorff system as previously described (Stenslokken *et al*, 2009). Hearts were stabilized for 20 min before being subjected to 25 min global ischemia and 30 min of reperfusion. In some hearts, xestospongin C was added to the perfusion buffer at a final concentration of 2 μmol/l during the pre-ischemic stabilization period. Coronary effluents were collected for troponin I measurements. Hearts were snap-frozen after 30 min reperfusion for subsequent immunoblotting analyses. Cardiac troponin I was measured using a high sensitivity mouse cardiac troponin I ELISA kit (Life Diagnostics Inc., West Chester, PA, USA).

## Statistics

Data are presented as mean ± SEM. For *in vitro* experiments, a typical sample size of 3–6/group was used. For imaging, at least 100

different cells/group were used. For *ex vivo* studies, we used 6–7 animals/group. Comparisons among groups were undertaken by Student's *t*-test, one-way, or two-way ANOVA using GraphPad Prism 8.3 (GraphPad, San Diego, CA, USA) and the recommended post hoc test method (Tukey's or Dunnett's test).

## Data availability

There are no primary datasets or computer codes associated with the study.

**Expanded View** for this article is available online.

### Acknowledgments
This work was supported by grants from the British Heart Foundation (RG/13/11/30384, CH/1999001/11735) and a Fondation Leducq Transatlantic Network of Excellence award (17CVD04) to AMS, and in part by the Department of Health via a National Institute for Health Research (NIHR) Biomedical Research Centre award to Guy's & St Thomas' NHS Foundation Trust in partnership with King's College London. VVB was supported by the Russian Science Foundation (17-14-01086) and DFG IRTG 1816. CCJM was supported by Arthritis Research UK grant ARUK-EG2013B-1. KS-B and KS are supported by the DZHK (German Center for Cardiovascular Research) Partner Sites Goettingen and Rhein-Main respectively. We thank the Wohl Cellular Imaging Centre at King's College London, especially Dr. George Chennell, for help with microscopy.

### Author contributions
MB performed experiments, analyzed data, and co-wrote the manuscript. CXCS, CM, ADH, and KB performed experiments. CCJM, RAF, and MP supervised specific experiments. AR, KS, KS-B, JHD, and VVB generated and provided key reagents. T-PS provided conceptual input. AMS designed and supervised the study and wrote the manuscript.

### Conflict of interest
The authors declare that they have no conflict of interest.

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
