## [Review Process File · The EMBO Journal]

Nox4 regulates InsP₃ receptor-dependent Ca²⁺ release into mitochondria to promote cell survival

Matteo Beretta, Celio Santos, Chris Molenaar, Anne Hafstad, Christopher Miller, Aram Revazian, Kai Betteridge, Katrin Schröder, Katrin Streckfuß-Bömeke, James Doroshov, Roland Fleck, Tsung-Ping Su, Vsevelod Belousov, Maddy Parsons, and Ajay Shah

DOI: [10.15252/embj.2019103530](https://doi.org/10.15252/embj.2019103530)

Corresponding author(s): Ajay Shah (ajay.shah@kcl.ac.uk)

Review Timeline:

Submission Date:	22nd Sep 19
Editorial Decision:	28th Oct 19
Revision Received:	11th Feb 20
Editorial Decision:	20th Mar 20
Revision Received:	19th Apr 20
Editorial Decision:	19th May 20
Revision Received:	22nd Jun 20
Accepted:	1st Jul 20

Editor: Elisabetta Argenzio

Transaction Report:

Dear Dr. Shah,

Thank you for submitting your manuscript entitled "Nox4 regulates InsP3 receptor Ca²⁺ release to mitochondria to promote cell survival" to The EMBO Journal. Your study has been sent to three referees for evaluation, whose reports are enclosed below.

As you can see, while the referees find the work potentially interesting, they also raise several key points that need to be addressed before they can support publication in The EMBO Journal. We find that all three reports are fair and balanced and require you to add new experimental data and controls, as well as appropriate quantifications and statistical analyses, in order to satisfy every request from the referees.

Given the overall interest of your study, I would like to invite you to revise the manuscript in response to the referee requests. I should also note that conclusively addressing all major and minor issues raised by the referees would be essential for publication in The EMBO Journal, as well as a strong support from the reviewers.

When preparing your letter of response to the referees' comments, bear in mind that this will form part of the Review Process File and will be available online to the community. For more details on our Transparent Editorial Process, please visit our website:
http://emboj.embopress.org/about#Transparent_Process

We generally grant three months as standard revision time. As a matter of policy, competing manuscripts published during this period will not negatively impact on our assessment of the conceptual advance presented by your study. Nevertheless, please contact me as soon as possible upon publication of any related work.

I thank you again for the opportunity to consider this study for publication and will be happy to answer any questions about the submission of the revised manuscript to The EMBO Journal. I look forward to your revision.

Yours sincerely,

Elisabetta Argenzio, PhD
Editor
The EMBO Journal

- a point-by-point response to the referees' comments, with a detailed description of the changes made (as a word file).

- a word file of the manuscript text.

- individual production quality figure files (one file per figure)

- a complete author checklist, which you can download from our author guidelines (<https://www.embopress.org/page/journal/14602075/authorguide>).

- Expanded View files (replacing Supplementary Information)

Further information is available in our Guide For Authors:

The revision must be submitted online within 90 days; please click on the link below to submit the revision online before 26th Jan 2020.

Link Not Available

Referee #1:

In this manuscript the authors studied on the mechanism by which Nox4 protect cells from regulated cell death. Based on their findings the authors conclude that Nox4 is localized at the MAM where it generates ROS at the specific and restricted compartment of the MAM. ROS serves to inhibit PP2a to increase the level of p-Akt and the activated kinase phosphorylates and inhibits the activity of IP3 receptors at the MAM. This in turn restricts Ca²⁺ influx into the mitochondria to prevent opening of the mitochondrial permeability transition pore and cell death.

The topic is of high interest and the finding that Nox4 functions at the MAM to control mitochondrial Ca²⁺ load through phosphorylation of IP3 receptors and cell death is significant. In general, the findings support the major conclusion but stop short of fully establishing the functioning of the pathway under rest and stimulated conditions. This should be examined by additional experiments before further considerations.

1. Figure 2: The authors should use the cardiomyocytes in panel A to determine the effect of SR Ca²⁺ release by receptor stimulation and by caffeine to determine whether knockdown of Nox4 affect mitochondrial Ca²⁺ influx in cardiomyocytes.
2. Figure 2I: It is not clear why the Nox4 mutant partially restores mitochondrial Ca²⁺ influx and some of the other Nox4 effect. One possibility is the Nox4 also functions as a tether at the MAM. This should be examined by determining whether expression of Nox4 and its inactive mutant in Nox4 knockout cells increases the number of MAM. This can be done by EM (Figure 3) and/or proximity assay and/or coimmunoprecipitation of ER and mitochondrial proteins in the MAM.
3. Figure 3 A-C and Figure 5 A, B: It is important to show whether cell stimulation affect the level of Nox4 in the MAM either using the same fractionation in stimulated cells and determining the effect of cell stimulation on coimmunoprecipitation of Nox4 with FACL4 and the IP3 receptors.
4. The results in Figure 3D should be quantified in multiple images.
5. The experiments in Figure 4 should be performed in resting and stimulated cells. The experiments in comments 3-5 should determine if cell stimulation further recruits Nox4 to the MAM.
6. It is surprising that the important results in Figure S5 were relegated to the supplement. This Figure, or at least panels B-D, F should be moved to the main text.

Referee #2:

The paper by Beretta et al. describes a new role for NOX4 in the regulation of cell death. They showed that NOX4 is located at Mitochondria-associated Membranes (MAMs), where it regulates the phosphorylation status of IP3Rs and Ca²⁺ efflux in an Akt-dependent manner, by modulating the amount of H₂O₂ exclusively at MAMs. The relevance of this molecular pathway is confirmed by *in vivo* experiments upon I/R injury. The authors conclude that this NOX4-dependent stress pathway protects against necrosis through spatially delimited redox signaling at the MAM to inhibit the Ca²⁺ release activity of IP3R.

Although the paper is new and interesting, several serious issues undermine confidence in these conclusions.

NOX4 depletion increased Ca²⁺ release from the ER with consequent high mitochondrial Ca²⁺ accumulation (Fig. 2). However, no differences in cytosolic Ca²⁺ levels have been detected. This is not easy to understand since enhanced Ca²⁺ efflux from the ER should result in higher cytosolic [Ca²⁺]. How can the authors account for such divergent findings?

In Figs 5-6, NOX4-KO cells displayed very low levels of phospho-IP3R, but the aminoacidic residue that is targeted by the phosphorylation event is not indicated. M&M section reported a phospho-IP3R antibody purchased by Cell Signalling Tech. Searching this Ab in the CST website, I found only one phospho-IP3R antibody, which recognizes SER 1756. Is the same Ab used in the manuscript? If so, some problems exist i) cAMP-dependent protein kinase (PKA), not Akt, is the kinase responsible for the phosphorylation at Ser1756. Akt phosphorylates IP3Rs at the C-term; ii) Phosphorylation on Ser1756 is specific for type 1 IP3R since IP3R2 and IP3R3 lack this consensus PKA phosphorylation site. Of note, the authors have analyzed only IP3R3, which is enriched at MAMs; iii) Additional studies established that PKA phosphorylation of IP3R1 increases, rather than decreases, IP3-induced Ca²⁺ release by approximately 2-fold (Nakade S. et al., JBC 1994; Gomez L. et al., CDDis

2016). Together, all these issues arise doubts on the molecular mechanism presented here. In Fig. 5, NOX4 depletion abolished ROS production only at MAMs but not in the cytosol. However, H₂O₂ freely and rapidly pass the membranes to diffuse inside the whole cell; thus I expected a reduction of H₂O₂ levels also in the cytosol. In Fig. 3A-B-C MAMs fractions appear not enriched for typical MAM markers FACL4, SIGMA 1R and calnexin

Referee #3:

The manuscript by Beretta et al. analyzes the protective role of Nox4 in counteracting necrotic cell death upon stress -either serum starvation in cells or I/R injury in hearts. The authors propose that NOX-dependent production of ROS at MAMs is required for Akt activation, InsP3R phosphorylation and, as a consequence, for ER-mitochondria Calcium transfer reduction which, in stress conditions, would prevent mitochondrial calcium overload, loss of mitochondrial membrane potential (MMP) and cell death.

Major:

In general, the study is quite well designed and experiments are rigorously performed, however, I have one major concern that must be addressed. The authors show that Ad-shNox4 or Nox4KO cells when serum-starved for 48 hours are subjected to a) cell death, b) loss of MMP (figure 1), but still their mitochondria take up more calcium than wt controls when treated with an InsP3-generating agonist (figure 2). This is against the thermodynamics properties of mitochondrial calcium uptake, which occurs only in energized mitochondria and strictly depends on the MMP. The notion of mitochondrial calcium overload, which may occur upon pathological triggers and eventually result in PTP-dependent cell death, differs from that of mitochondrial calcium uptake that, as said, occurs only in energized mitochondria. Thus, the mitochondrial calcium traces of figure 2 could have been recorded only from cells with polarized mitochondria and not from a depolarized cell population. To clarify this point, the authors should simultaneously detect TMRE-positive and mitochondria-targeted Cameleon positive cells, and distinguish agonist-induced mitochondrial calcium uptake traces of TMRE-positive from TMRE-negative cells.

Other major concerns:

On top of this, since the authors claim that inhibition of the PTP by CsA treatment is protective against serum depletion-induced death of Nox4KO cells, mitochondrial calcium traces in TMRE+ and TMRE- cells should be additionally performed in the presence or absence of CsA.

Is the mitochondrial calcium uptake increase upon Nox4 deletion due to decreased ROS production? Measurements of mitochondrial calcium uptake in catalase treated cells (incubation time with catalase according to fig 2I and 5C) should be performed.

The authors show that Nox4 is required for ROS production at MAMs. However, in view of the involvement of mitochondria in the process and the important role played by mitochondria in ROS production, the authors should analyze whether mitochondrial-derived ROS are also involved. For this purpose, in serum-starved wt cells, mitochondrial ROS should be measured by means of the mitoHyPer probe. In addition, they should analyze the effects of a mitochondria-targeted antioxidant (e.g. MitoTEMPO) on a) phosphorylation of Akt and InsP3R; b) H₂O₂ measured by FACL4Hyper; c) mitochondrial calcium uptake.

In figure 6, the role of InsP3R by XeC treatment is well determined. Less clear is the role of Akt. Indeed, the use of an Akt inhibitor on top of Nox4 deletion is not very informative. The authors should show that increased Akt activity (e.g. by overexpression of a constitutively active Akt expression vector or a similar strategy) rescues mitochondrial calcium levels and cell viability in

Nox4KO cells.

Minor:

Why are the effects of Nox4 deletion evident only in stress conditions? Nox4 protein is present also in 15% serum although at lower levels compared to serum-free conditions. Which is the activity level of Nox4 in serum-repleted conditions? ROS (particularly MAM-localized ROS) should be measured in serum-repleted conditions in the presence/absence of Nox4.

Figure S1B: what is represented here compared to fig. 1C?

Figure S4: does catalase increase Ser/Thr phosphatase activity similarly to Nox4 deletion? Is phosphatase activity rescued in Nox4KO cells upon wtNox4 or Nox4mut expression?

Beretta et al: Nox4 regulates InsP₃ receptor Ca²⁺ release to mitochondria to promote cell survival. EMBOJ 2019_103530

We thank the referees for the positive reviews and the helpful comments and suggestions all of which we hope we have addressed in this revision. A point-by-point response is provided below.

Referee #1:

1. Figure 2: The authors should use the cardiomyocytes in panel A to determine the effect of SR Ca²⁺ release by receptor stimulation and by caffeine to determine whether knockdown of Nox4 affect mitochondrial Ca²⁺ influx in cardiomyocytes.

We performed this experiment and the results are shown in new Fig EV2 D and described on page 8 para 1. They show that an increased mitochondrial calcium influx in cardiomyocytes after Nox4 knockdown is observed after histamine but not caffeine stimulation. This is consistent with the notion that Nox4 specifically modulates flux through InsP₃R at the MAM and does not directly affect ER/SR calcium release through other channels (RyR).

2. Figure 2I: It is not clear why the Nox4 mutant partially restores mitochondrial Ca²⁺ influx and some of the other Nox4 effect. One possibility is the Nox4 also functions as a tether at the MAM. This should be examined by determining whether expression of Nox4 and its inactive mutant in Nox4 knockout cells increases the number of MAM. This can be done by EM (Figure 3) and/or proximity assay and/or coimmunoprecipitation of ER and mitochondrial proteins in the MAM.

We performed proximity ligation assays (PLA) in Nox4 KO cells transfected with functional Nox4 or Nox4^{P437H}. These studies showed that there was no change in the MAM as assessed by the FAFL4/InsP₃R signal. The results are shown in Appendix Fig S4H and described on page 10 para 1.

3. Figure 3 A-C and Figure 5 A, B: It is important to show whether cell stimulation affect the level of Nox4 in the MAM either using the same fractionation in stimulated cells and determining the effect of cell stimulation on coimmunoprecipitation of Nox4 with FAFL4 and the IP3 receptors.

We now show that Nox4 levels at the MAM are increased after serum starvation – new Fig EV3 and Results Page 9 para 1.

4. The results in Figure 3D should be quantified in multiple images.

We quantified from multiple images the proportion of Nox4 or FAFL4 labeled at the MAM versus any other location in the cell. The results are shown in new Fig S3 and confirm a predominant MAM location.

5. The experiments in Figure 4 should be performed in resting and stimulated cells. The experiments in comments 3-5 should determine if cell stimulation further recruits Nox4 to the MAM.

This new experiment is shown in Fig EV3 B and reported on page 9 last para. It confirms an increase in Nox4 at the MAM after serum starvation. Please also see response to point 3 above.

6. It is surprising that the important results in Figure S5 were relegated to the supplement. This Figure, or at least panels B-D, F should be moved to the main text.

We have now placed these results in the new Expanded View Figure EV5 which forms part of the main manuscript.

Referee #2:

NOX4 depletion increased Ca²⁺ release from the ER with consequent high mitochondrial Ca²⁺ accumulation (Fig. 2). However, no differences in cytosolic Ca²⁺ levels have been detected. This is not easy to understand since enhanced Ca²⁺ efflux from the ER should result in higher cytosolic [Ca²⁺]. How can the authors account for such divergent findings?

Our results suggest that Nox4 modulates efflux through InsP₃R specifically at the MAM (because Nox4 is located there). Therefore, it only modulates that part of the calcium released from the ER that is released at the MAM. We do see increases in cytosolic calcium after histamine or ATP (Fig 2G-H) but these are not significantly affected by Nox4 because it only affects the calcium release at the MAM. This is further supported by the new experiments performed in response to point 1 from reviewer 1 (new Fig EV2 D) which show that caffeine-induced calcium release in cardiomyocytes is not affected by Nox4.

In Figs 5-6, NOX4-KO cells displayed very low levels of phospho-IP3R, but the aminoacidic residue that is targeted by the phosphorylation event is not indicated. M&M section reported a phospho-IP3R antibody purchased by Cell Signalling Tech. Searching this Ab in the CST website, I found only one phospho-IP3R antibody, which recognizes SER 1756. Is the same Ab used in the manuscript? If so, some problems exist i) cAMP-dependent protein kinase (PKA), not Akt, is the kinase responsible for the phosphorylation at Ser1756. Akt phosphorylates IP3Rs at the C-term; ii) Phosphorylation on Ser1756 is specific for type 1 IP3R since IP3R2 and IP3R3 lack this consensus PKA phosphorylation site. Of note, the authors have analyzed only IP3R3, which is enriched at MAMs; iii) Additional studies established that PKA phosphorylation of IP3R1 increases, rather than decreases, IP3-induced Ca²⁺ release by approximately 2-fold (Nakade S. et al., JBC 1994; Gomez L. et al., CDDis 2016). Together, all these issues arise doubts on the molecular mechanism presented here.

The referee raises an important point. We previously used an Ab which we believe detected Akt-mediated phosphorylation based on the results of complete inhibition of InsP₃R phosphorylation by Akti. However, we have now repeated these experiments by studying InsP₃R phosphorylation with a different approach. We have now immunoprecipitated InsP₃R and then immunoblotted with a monoclonal Ab against the phosphorylated Akt-substrate motif RXRXX(pS/T) (Cell Signaling 23C8D2; mAb #10001) as in Khan et al., 2006 – please see Appendix Fig S5A. The overall results of these studies are similar to those in the previous version.

In Fig. 5, NOX4 depletion abolished ROS production only at MAMs but not in the cytosol. However, H₂O₂ freely and rapidly pass the membranes to diffuse inside the whole cell; thus I expected a reduction of H₂O₂ levels also in the cytosol.

Actually, specific redox signaling generally depends critically upon spatially confined ROS signals which are achieved through several mechanisms including (a) localised low level ROS production in subcellular compartments and (b) very high cellular capacity to degrade ROS through molecules such as GSH and peroxiredoxin. This means that even though H₂O₂ is considered to be highly diffusible, in practice its effects are restricted close to the site of endogenous production unless the level of production is so high that it overwhelms cellular antioxidant systems. Therefore, it is not surprising that Nox4-dependent ROS production at the MAM does not affect ROS levels in other cellular locations. For detailed reviews on this topic please see, for example, D'Autre'aux & Toledano, 2007; Winterbourn 2013.

D'Autre'aux B, Toledano MB (2007) ROS as signalling molecules: mechanisms that generate specificity in ROS homeostasis. Nat Rev Mol Cell Biol 8:813 – 824.

Winterbourn CC (2013) The biological chemistry of hydrogen peroxide. Methods Enzymol. 528:3-25.

In Fig. 3A-B-C MAMs fractions appear not enriched for typical MAM markers FACL4, SIGMA 1R and calnexin

As far as we can see, we think these figures show an enrichment of the MAM markers (VDAC, calnexin, Sigma1R, FACL4) in the MAM fraction (lane 6) and in the crude ER fraction which includes MAM (lane 3), unless we are missing something.

Referee #3:

Major:

In general, the study is quite well designed and experiments are rigorously performed, however, I have one major concern that must be addressed. The authors show that Ad-shNox4 or Nox4KO cells when serum-starved for 48 hours are subjected to a) cell death, b) loss of MMP (figure 1), but still their mitochondria take up more calcium than wt controls when treated with an InsP3-generating agonist (figure 2). This is against the thermodynamics properties of mitochondrial calcium uptake, which occurs only in energized mitochondria and strictly depends on the MMP. The notion of mitochondrial calcium overload, which may occur upon pathological triggers and eventually result in PTP-dependent cell death, differs from that of mitochondrial calcium uptake that, as said, occurs only in energized mitochondria. Thus, the mitochondrial calcium traces of figure 2 could have been recorded only from cells with polarized mitochondria and not from a depolarized cell population. To clarify this point, the authors should simultaneously detect TMRE-positive and mitochondria-targeted Cameleon positive cells, and distinguish agonist-induced mitochondrial calcium uptake traces of TMRE-positive from TMRE-negative cells.

We have performed the experiment suggested by the referee (new Fig EV2 A-B). This clearly shows that the enhanced histamine-induced mitochondrial calcium uptake in Nox4-deficient cells is observed only in TMRE+ cells (panel A) and not in TMRE- cells (panel B). These results are now described on page 8 para 1. Please note that the calcium experiments were done after 24 h serum starvation which is now clarified in the methods (page 22).

On top of this, since the authors claim that inhibition of the PTP by CsA treatment is protective against serum depletion-induced death of Nox4KO cells, mitochondrial calcium traces in TMRE+ and TMRE- cells should be additionally performed in the presence or absence of CsA.

We checked if the increased mitochondrial uptake in Nox4-deficient cells involved the mPTP by performing experiments in the presence or absence of bongrekinic acid. We used this inhibitor rather than cyclosporin A because the latter is reported to also inhibit calcium uptake independent of the mPTP (Montero et al., 2004). Bongrekinic acid had no effect on mitochondrial calcium uptake (new Fig EV2 C and Results page 8 para 1) but it did inhibit cell death (new Fig S1E)

Montero, M., et al., (2004) Calcineurin-independent inhibition of mitochondrial Ca²⁺ uptake by cyclosporin A. *Br J Pharmacol* 141, 263-268.

Is the mitochondrial calcium uptake increase upon Nox4 deletion due to decreased ROS production? Measurements of mitochondrial calcium uptake in catalase treated cells (incubation time with catalase according to fig 2I and 5C) should be performed.

This experiment is shown in Fig 2I (third column). Catalase treatment does indeed result in an increase in mitochondrial calcium uptake.

The authors show that Nox4 is required for ROS production at MAMs. However, in view of the involvement

of mitochondria in the process and the important role played by mitochondria in ROS production, the authors should analyze whether mitochondrial-derived ROS are also involved. For this purpose, in serum-starved wt cells, mitochondrial ROS should be measured by means of the mitoHyPer probe. In addition, they should analyze the effects of a mitochondria-targeted antioxidant (e.g. MitoTEMPO) on a) phosphorylation of Akt and InsP₃R; b) H₂O₂ measured by FACLSHyper; c) mitochondrial calcium uptake.

As suggested, we assessed ROS levels in the mitochondria using mito-targeted HyPer probes but found no difference between WT and Nox4 KO cells (new Fig EV4 A). The treatment of WT cells with Mito-TEMPO had no effect on the level of phosphorylation of Akt or InsP₃R (new Fig EV4 B). Mito-TEMPO also had no significant effect on the histamine-induced increase in mitochondrial calcium uptake (new Fig EV 4C). These results suggest that mitochondrial ROS do not contribute significantly to the effects reported in this study. These data are described in the Results page 11, para 2.

In figure 6, the role of InsP₃R by XeC treatment is well determined. Less clear is the role of Akt. Indeed, the use of an Akt inhibitor on top of Nox4 deletion is not very informative. The authors should show that increased Akt activity (e.g. by overexpression of a constitutively active Akt expression vector or a similar strategy) rescues mitochondrial calcium levels and cell viability in Nox4KO cells.

We have performed new experiments using okadaic acid to inhibit phosphatase activity and thereby enhance Akt activation. Okadaic acid significantly increased both Akt and InsP₃R phosphorylation in Nox4 KO cells (new Appendix Fig S6B). Okadaic acid also reduced the histamine-induced increase in mitochondrial calcium levels and cell death in serum-starved Nox4 KO cells (new Fig EV5 D-E). These data are presented in the Results page 12 second para.

Minor:

Why are the effects of Nox4 deletion evident only in stress conditions? Nox4 protein is present also in 15% serum although at lower levels compared to serum-free conditions. Which is the activity level of Nox4 in serum-repleted conditions? ROS (particularly MAM-localized ROS) should be measured in serum-repleted conditions in the presence/absence of Nox4.

We assessed the MAM-localized ROS signal under serum-replete conditions and the results are shown in Appendix Fig S5C and reported on page 11, para 2. We observed a higher MAM-located ROS signal in WT versus Nox4 KO cells but the magnitude was lower than under serum-deficient conditions. The fact that functional effects are only observed under stress conditions is likely to be because: (a) the target for Nox4 effects (i.e. Akt activation and InsP₃R phosphorylation) is activated in response to stress (b) the amount of Nox4 and Nox4-mediated ROS production goes up during stress.

Figure S1B: what is represented here compared to fig. 1C?

This showed essentially the same information but using trypan blue exclusion. We have now deleted this panel.

Figure S4: does catalase increase Ser/Thr phosphatase activity similarly to Nox4 deletion? Is phosphatase activity rescued in Nox4KO cells upon wtNox4 or Nox4mut expression?

We performed additional experiments to test this question. The results (new Fig EV4E and page 12 para 1) show that incubation of serum-starved WT cells with PEG-catalase resulted in a significant increase in phosphatase activity. In Nox4 KO cells transfected with functional Nox4, the serine/threonine phosphatase activity was decreased whereas transfection with Nox4P437H had no significant effect.

Dear Dr. Shah,

Thank you for submitting a revised version of your manuscript. It has now been seen by the original referees, whose comments are shown below.

As you will see, while referee #1 and 3 find that their criticisms have been sufficiently addressed and recommend the manuscript for publication, reviewer #2 feels that his/her doubts on the molecular mechanism persist. In addition, s/he points to inconsistencies in the results.

We agree with referee #2 that these are important points that have to be addressed before pursuing publication of this manuscript and would thus invite you to address these remaining issues.

I look forward to your revision.

Yours sincerely,

Elisabetta Argenzio, PhD
Editor
The EMBO Journal

When assembling figures, please refer to our figure preparation guideline in order to ensure proper formatting and readability in print as well as on screen:
<http://bit.ly/EMBOPressFigurePreparationGuideline>

The revision must be submitted online within 90 days; please click on the link below to submit the revision online before 18th Jun 2020.

Link Not Available

Referee #1:

The authors addressed all of my concerns and the m/s is ready for publication in the EMBO journal.

Referee #2:

Although the new version of the manuscript by Beretta et al. appears ameliorated, the authors' reply to my queries failed to convince me.

In particular, the authors state that they used a new Ab for the recognition of Akt-mediated phosphorylation on IP3R. However, the phosphor-Akt substrates antibody can be used only upon immunoprecipitation of the target protein. So, it is not easy to understand the results showed in Figs. 5C, 6F and Appendix S5B, which are annotated as "Immunoblots for phosphorylated and total Akt and InsP3R in crude mitochondrial fractions", and not as IP exps. Again, the data in Appendix Fig. S5A simply demonstrated that in basal condition, the IP3R (type 3?) is phosphorylated by Akt. Nevertheless, the authors do not clarify the Ab used in the first version of the paper. If it is the phosphor-IP3R1 (Ser1756) antibody (PKA-dependent phosphorylation), the previous data cannot be ignored, and the role of PKA should be considered. Therefore, all my doubts about the molecular mechanism still persist.

In fig. 2E-F the authors showed that in NOX4 KO Mefs displayed increased total Ca²⁺ release from the ER through the IP3R. This MUST reflect a higher cytosolic Ca²⁺ increase, not only mitochondrial (unless there are some parallel compensatory mechanisms inside the cytosol). Thus, the idea that Nox4 only affects the calcium release at the MAM is at odds with the alteration of global ER Ca²⁺ efflux.

Referee #3:

The authirs responded to all my questions

Beretta et al. Nox4 regulates InsP₃ receptor Ca²⁺ release to mitochondria to promote cell survival.

Response to Referee #2

Thank you for these further comments. A point by point response and a description of the changes made to the manuscript is provided below. We hope that this addresses the main issues and the paper is now considered suitable for publication.

Although the new version of the manuscript by Beretta et al. appears ameliorated, the authors' reply to my queries failed to convince me.

In particular, the authors state that they used a new Ab for the recognition of Akt-mediated phosphorylation on IP3R. However, the phosphor-Akt substrates antibody can be used only upon immunoprecipitation of the target protein. So, it is not easy to understand the results showed in Figs. 5C, 6F and Appendix S5B, which are annotated as "Immunoblots for phosphorylated and total Akt and InsP3R in crude mitochondrial fractions", and not as IP exps. Again, the data in Appendix Fig. S5A simply demonstrated that in basal condition, the IP3R (type 3?) is phosphorylated by Akt.

Response: We did actually perform immunoprecipitation for InsP₃R-type 3 and then immunoblotting for the phosphorylated Akt-substrate motif RXRXX(pS/T) in all the above experiments, exactly as the reviewer indicates should be done. This was stated both in the revised text and the Methods. However, we have now further emphasised this point by (1) stating this in the legends to new Figs 5C, 6F EV4 B and Appendix S5B; (2) annotating these figures as shown in the example below to make this very clear; (3) making the description in the Methods (**page 20, bottom**) even more explicit. [Figures for referees not shown.]

Nevertheless, the authors do not clarify the Ab used in the first version of the paper. If it is the phosphor-IP3R1 (Ser1756) antibody (PKA-dependent phosphorylation), the previous data cannot be ignored, and the role of PKA should be considered. Therefore, all my doubts about the molecular mechanism still persist.

Response: The previous Ab used based on recommendation by a colleague was from Cell Signaling (#8548, stated in the catalogue to be targeted against InsP₃R-type 1 phosphorylated at Ser1756). However, as noted by this reviewer, we are investigating the InsP₃R-type 3, which is enriched at the MAM and does not have the Ser1756 phosphosite. In the previous data, the InsP₃R phosphorylation was very efficiently blocked by Akti (**old Fig S6A**), so we believe the most likely explanation is that the Ab may be detecting other phosphosites. More importantly, the new results with the Ab targeted against the phosphorylated Akt-substrate motif RXRXX(pS/T) are very clear and robust, and completely consistent with the mechanism we are proposing.

Turning to any possible role of PKA, we would like to point out that (1) InsP₃-type3 does not have the Ser1756 PKA phosphosite and (2) previous work in the literature shows that PKA does not significantly modulate the type 3 receptor (Taylor et al. Cell Calcium 2017;63:48-52; Soulsby et al. Cell Calcium 2007;42:261-70). In the study by Soulsby et al., they also showed that mutation of other potential PKA target sites in InsP₃R-type 3 receptors had no effect on Ca²⁺ flux through the channel. Therefore, we believe it is very unlikely that there is a significant role for PKA.

We have now added a note regarding the lack of effect of PKA on InsP₃R-type 3 receptors in the Discussion (**page 15, last para**) and also included the reference by Soulsby et al.

In fig. 2E-F the authors showed that in NOX4 KO Mefs displayed increased total Ca²⁺ release from the ER through the IP3R. This MUST reflect a higher cytosolic Ca²⁺ increase, not only mitochondrial (unless there are some parallel compensatory mechanisms inside the cytosol). Thus, the idea that Nox4 only affects the calcium release at the MAM is at odds with the alteration of global ER Ca²⁺ efflux.

Response: The reviewer makes a fair point that if there is a significantly greater decrease in ER Ca²⁺ content in Nox4 KO cells after application of an agonist that opens InsP₃ channels, then one would normally expect a corresponding increase in cytosolic Ca²⁺ signal since the amount of flux purely at the MAM might not be high enough to markedly reduce ER Ca²⁺ content. In fact, when histamine is used as the agonist (Fig 2E and G), there is a very small decrease in ER Ca²⁺ in Nox4 KO cf. WT and a trend to a slight increase in cytosolic Ca²⁺ - so there is no real inconsistency in this experiment. The potential discrepancy is observed with ATP, where there is a larger decrease (~ 8%) in ER Ca²⁺ in Nox4 KO cf. WT but no change in the cytosolic Ca²⁺. We do not have an obvious explanation for this apart from the reviewer's suggestion that there may be some parallel compensatory mechanism in the cytosol when ATP is used. We have now altered the description of these results on **page 7**, highlighting the difference between the agonists and stating that additional mechanisms may be at play in the case of ATP – please see text below.

"The results with histamine are consistent with enhanced calcium transfer from the ER to mitochondria in the Nox4 KO cells. In the case of ATP, the greater depletion of ER calcium in Nox4

KO versus WT cells suggests that there may be additional effects. This magnitude of difference in ER calcium would be expected to be accompanied by differences in cytosolic calcium transients, so there may be other compensatory mechanisms at play with this agonist.”

Dear Ajay,

Thank you for submitting a revised version of your manuscript. Please accept my apologies for the delay in getting back to you with our decision. The manuscript has now been seen by referee #2, whose comments are shown below.

As you will see, while this referee finds the role of NOX4 in regulating ER Ca²⁺ homeostasis still doubtful, s/he has no additional concerns. Also, referee #2 suggests that you rearrange some figures. I would thus invite you to tone down all conclusions on how NOX4 regulates ER Ca²⁺ homeostasis. In addition, there are a few editorial issues concerning text and figures that I need you to address before we can officially accept the manuscript.

-> Please indicate the number of replicas used for calculating statistics.

-> Include a "Data Availability" section (placed after Material and Methods section) even if there are no primary datasets and computer codes associated with the study.

-> See our author guidelines for more detail <http://emboj.embopress.org/authorguide>

-> Anti-Nox4 in Fig EV3A and anti-Akt in Fig S5B do not match source data. Please clarify.

-> Provide source data images at higher resolution.

-> Provide one source data file for each figure. Add boxes to source data to highlight the corresponding area in the main figure.

-> Our production/data editors have asked you to clarify several points in the figure legends (see attached document). Please incorporate these changes in the attached word document and return it with track changes activated.

-> Papers published in The EMBO Journal include a 'Synopsis' to further enhance discoverability. Synopses are displayed on the html version of the paper and are freely accessible to all readers. The synopsis includes a short standfirst as well as 2-5 one sentence bullet points that summarise the paper and are provided by the authors. I would therefore ask you to include your suggestions for bullet points. Please note that synopsis blurb and bullet points can be edited by the editors.

-> In addition, I would encourage you to provide an image for the synopsis. This image should provide a rapid overview of the question addressed in the study but still needs to be kept fairly modest since the image size cannot exceed 550x400 pixels.

Thank you again for giving us the chance to consider your manuscript for The EMBO Journal, I look forward to your revision.

Best regards,

Elisabetta

Elisabetta Argenzio, PhD
Editor
The EMBO Journal

Further information is available in our Guide For Authors:

The revision must be submitted online within 90 days; please click on the link below to submit the revision online before 17th Aug 2020.

Referee #2:

My suggestions are to move Figs. 2G-H to supplementary items and to expand the discussion. Indeed, the analysis of cytosolic $[Ca^{2+}]$ is not a key finding in this paper, which is mainly focused on ER-mitochondria dynamics. However, based on the cytosolic Ca^{2+} measurements presented here, some doubts remain on the role of NOX4 in regulating ER Ca^{2+} homeostasis.

EMBOJ-2019-103530

Beretta et al., Nox4 regulates InsP₃ receptor Ca²⁺ release to mitochondria to promote cell survival

Editor's comment

As you will see, while this referee finds the role of NOX4 in regulating ER Ca²⁺ homeostasis still doubtful, s/he has no additional concerns. Also, referee #2 suggests that you rearrange some figures. I would thus invite you to tone down all conclusions on how NOX4 regulates ER Ca²⁺ homeostasis.

Response

We have moved Figs. 2G-H to Appendix Fig S2 (panels C, D) as requested.

We have also toned down all the parts of the manuscript dealing with NOX4 and cytosolic calcium (changes are tracked in the paper).

Editorial issues

-> Please indicate the number of replicas used for calculating statistics.

This has been indicated in the legends to each figure and in the Methods.

-> Include a "Data Availability" section (placed after Material and Methods section) even if there are no primary datasets and computer codes associated with the study.

Now included.

-> Anti-Nox4 in Fig EV3A and anti-Akt in Fig S5B do not match source data. Please clarify.

Fig EV3A – unfortunately the source blot was flipped over vertically during preparation and we also used a more exposed image. We sent by email a powerpoint showing the different exposures and the correctly flipped blot. The image with correct exposure and orientation is now included in the source blot file.

Fig S5B – the image used in the source file was a slightly different exposure from the one in the figure. We have now included the correct exposure image in the source file.

We also sent the Editor an email on 19.5.2020 with a Powerpoint file showing the different exposures for both the above figures so as to be completely transparent.

-> Provide source data images at higher resolution.

Now provided.

-> Provide one source data file for each figure. Add boxes to source data to highlight the corresponding area in the main figure.

Now done.

-> Our production/data editors have asked you to clarify several points in the figure legends (see attached document). Please incorporate these changes in the attached word document and return it with track changes activated.

Done.

-> Papers published in The EMBO Journal include a 'Synopsis' to further enhance discoverability. I would therefore ask you to include your suggestions for bullet points. Please note that synopsis blurb and bullet points can be edited by the editors.

Synopsis suggestions now included.

-> In addition, I would encourage you to provide an image for the synopsis. This image should provide a rapid overview of the question addressed in the study but still needs to be kept fairly modest since the image size cannot exceed 550x400 pixels.

An image for the synopsis is included.

Dear Ajay,

I am pleased to inform you that your manuscript has been accepted for publication in the EMBO Journal.

Congratulations!

Kind regards,

Elisabetta

Elisabetta Argenzio, PhD
Editor
The EMBO Journal

Please note that it is EMBO Journal policy for the transcript of the editorial process (containing referee reports and your response letter) to be published as an online supplement to each paper. If you do NOT want this, you will need to inform the Editorial Office via email immediately. More information is available here: http://emboj.embopress.org/about#Transparent_Process

Your manuscript will be processed for publication in the journal by EMBO Press. Manuscripts in the PDF and electronic editions of The EMBO Journal will be copy edited, and you will be provided with page proofs prior to publication. Please note that supplementary information is not included in the proofs.

Should you be planning a Press Release on your article, please get in contact with embojournal@wiley.com as early as possible, in order to coordinate publication and release dates.

If you have any questions, please do not hesitate to call or email the Editorial Office. Thank you for your contribution to The EMBO Journal.

** Click here to be directed to your login page: <http://emboj.msubmit.net>

Corresponding Author Name: Ajay Shah

Journal Submitted to: EMBO J

Manuscript Number: EMBOJ-2019-103530